# Ineq-Comp: Benchmarking Human-Intuitive Compositional Reasoning in Automated Theorem Proving on Inequalities

**Haoyu Zhao**[*†]   **Yihan Geng**[‡]   **Shange Tang**[†]   **Yong Lin**[†]   **Bohan Lyu**[§]   **Hongzhou Lin**[¶]

**Chi Jin**[*†]                    **Sanjeev Arora**[*†]

## Abstract

LLM-based formal proof assistants (e.g., in Lean) hold great promise for automating mathematical discovery. But beyond syntactic correctness, do these systems truly understand mathematical structure as humans do? We investigate this question in context of mathematical inequalities—specifically the prover's ability to recognize that the given problem simplifies by applying a known inequality such as AM/GM. Specifically, we are interested in their ability to do this in a *compositional setting* where multiple inequalities must be applied as part of a solution. We introduce Ineq-Comp, a benchmark built from elementary inequalities through systematic transformations, including variable duplication, algebraic rewriting, and multi-step composition. Although these problems remain easy for humans, we find that most provers—including Goedel, STP, and Kimina-7B—struggle significantly. DeepSeek-Prover-V2-7B shows relative robustness, but still suffers a 20% performance drop (pass@32). Even for DeepSeek-Prover-V2-671B model, the gap between compositional variants and seed problems exists, implying that simply scaling up the model size alone does not fully solve the compositional weakness. Strikingly, performance remains poor for all models even when formal proofs of the constituent parts are provided in context, revealing that the source of weakness is indeed in compositional reasoning. Our results expose a persisting gap between the generalization behavior of current AI provers and human mathematical intuition. All data and evaluation code can be found at https://github.com/haoyuzhao123/LeanIneqComp.

## 1   Introduction

Large language models (LLMs), like O3-mini (OpenAI, 2024) and Deepseek-R1 (Guo et al., 2025), have made huge progress in reasoning tasks. These models show outstanding performance in solving math problems, especially in natural language. However, reasoning using natural language is inherently unreliable for proofs, as subtle errors often arise in complex math derivations (Petrov et al., 2025). This shortcoming leads to increasing interest in applying LLMs to generate *formal math proofs*, where systems like Lean (De Moura et al., 2015; Moura and Ullrich, 2021), Isabelle (Paulson, 1994), and Coq (Barras et al., 1997), provide a framework for expressing and verifying proofs.

---

[*]Corresponding to: {haoyu,chij,arora}@princeton.edu

[†]Princeton Language and Intelligence, Princeton University.

[‡]Peking University.

[§]Tsinghua University

[¶]Amazon. This work is independent of and outside of the work at Amazon.

39th Conference on Neural Information Processing Systems (NeurIPS 2025) Track on Datasets and Benchmarks.

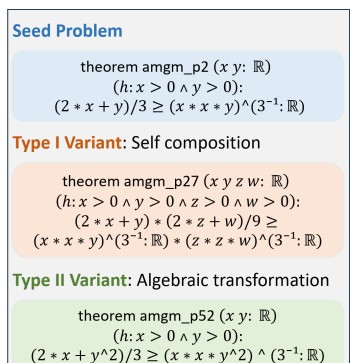
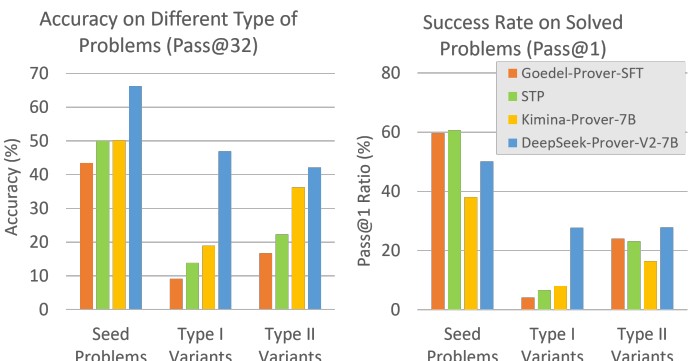

Figure 1: **Left:** Starting from a seed problem, we apply transformations that are intuitive to humans. Type I data are generated by duplicating the original inequality using distinct variable names and multiplying the two resulting inequalities. Type II data are created via algebraic manipulations. These transformed problems are trivial to solve if one has already understood the seed problem, and we would expect minimal performance drop between solving the seed problem and its transformed variations. **Mid:** Pass@32 accuracy of various models on the original seed problems and their transformed counterparts (Type I and Type II). All models, except DeepSeek-Prover-V2, exhibit substantial performance degradation on the transformed problems. Notably, even DeepSeek-Prover-V2-7B experiences a drop of over 20%. **Right:** Average success rate (Pass@1 within 3200 attempts) on the subset of problems each model is able to solve. While seed problems have relatively high solve rates across models, the transformed variants become significantly more challenging. For example, even for the strongest model—DeepSeek-Prover-V2-7B— its success rate drops from 50% on seed problems to 25% on their transformed counterparts. The drop suggests that current provers are sensitive to surface form rather than the underlying semantic equivalence of mathematical reasoning.

Parallel to progress in natural language reasoning, formal theorem proving with LLMs has developed rapidly in the past years. In particular, AlphaProof[6], AlphaGeometry (Trinh et al., 2024), and Seed-Prover (Chen et al., 2025) showed that a model could achieve silver-medal performance in the International Math Olympiad by finding verifiable proofs. Open-source research efforts also show significant progress through both whole-proof generation and tree-search approaches (Xin et al., 2024; Lin et al., 2025a; Dong and Ma, 2025; Wu et al., 2024; Li et al., 2024; Xin et al., 2025; Wang et al., 2025; Lin et al., 2025b), marking rapid advancements in formal theorem proving.

Despite rapid progress, evaluating a model's ability to generate formal proofs remains challenging due to lack of good benchmarks that can measure capabilities smoothly and comprehensively. The popular MiniF2F (Zheng et al., 2021) is small and yet its problems span a wide difficulty range from elementary to IMO level. ProofNet (Azerbayev et al., 2023) and PutnamBench (Tsoukalas et al., 2024) focus on difficult problems where current open models can only succeed on a few examples. Furthermore, both benchmarks are susceptible to data contamination.

We propose a new evaluation perspective: testing robustness to simple, human-intuitive transformations through *compositions*. We apply minor manipulations, like variable duplication or algebraic rewrites, to existing inequalities. These compositional variants are trivial for humans and even LLMs using natural language, yet they cause large performance drops in formal settings. This exposes brittleness in model reasoning that current benchmarks overlook.

The core of this failure is the lack of *compositional reasoning*. Complex arguments emerge by linking simpler, well-understood steps — echoing Newton's notion of "standing on the shoulders of giants" — which is an important skill in math reasoning. Following this insight, our benchmark leverages simple transformations that can be composed to create multi-step problems with hierarchical structures. This offers a natural and controllable axis for evaluating reasoning ability. Moreover, by systematically varying the depth and abstraction of these compositions, we can generate problem sequences with progressively increasing difficulty, allowing for more fine-grained and interpretable assessments than current benchmarks.

---

[6] https://deepmind.google/discover/blog/ai-solves-imo-problems-at-silver-medal-level/

Existing benchmarks do not explicitly test this "higher" understanding of compositional structure and their difficulty is often uneven or opaque (Zheng et al., 2021; Azerbayev et al., 2023; Tsoukalas et al., 2024; Wei et al.; Ren et al., 2025; Liu et al., 2025; Yu et al., 2025). Recent efforts like Yousefzadeh et al. (2025), which annotate IMO proofs with intermediate lemmas, follow a top-down approach that is hard to scale and may introduce subjective biases about valid solution paths. In contrast, our bottom-up construction composes seed problems into more complex ones through controllable transformations, enabling scalable and interpretable evaluation of formal reasoning.

Combining the above ideas, we introduce our new benchmark, Ineq-Comp, which is designed to assess the compositional reasoning abilities of formal theorem provers in algebraic inequalities. Using a bottom-up approach, we start from seed problems formalized in Lean 4, each solvable via standard inequalities such as AM-GM or Cauchy, and systematically apply simple, controlled transformations to generate more complex variants. These modifications are minimal and preserve human accessibility; in fact, general-purpose LLMs like OpenAI O3 solve them easily in informal natural language. Given that existing provers perform well on nontrivial problems in benchmarks like MiniF2F (Zheng et al., 2021) and ProofNet (Azerbayev et al., 2023), we expected them to generalize to these variants. Surprisingly, we found that most provers consistently fail to reason through these human intuitive transformations, exposing a fundamental weakness in their understanding of basic compositional generalization, which current benchmarks do not test for. (See Figure 1 for an example.) We summarize our contributions and key findings below:

1. We develop Ineq-Comp, a benchmark with 75 seed problems solved using basic tools such as AM-GM, Cauchy, and Jensen inequalities and accompanied by a verified Lean 4 proof, and 150 variants generated through compositional transformations (Figure 1 Left and Section 2.1). Ineq-Comp is designed to be fully extensible: further composed inequalities can be automatically generated using rule-based transformations on any seed set (Section 2.2). We also include 50 real-world inequality problems to enhance diversity (Section 2.3).

2. We benchmark a range of LLM-based provers—from general-purpose models like R1-Distill-Qwen to tree-search-based systems—on Ineq-Comp. All models, except DeepSeek-Prover-V2-7B (Ren et al., 2025), suffer from significant performance drops even on the simplest transformations (Section 3). Notably, DeepSeek-Prover-V2-7B also shows over 20% drop when tested under a limited computational budget (pass@32). These failures persist even when the model is scaled up (Section 4.1) or provided with the formal proof of the seed problem in-context (Section 4.2).

3. Through detailed output analysis, we identify two key limitations: (1) models rarely attempt to apply core inequalities such as AM-GM or Cauchy within their formal proofs, even when they refer to these techniques in natural language comments; and (2) most models rely heavily on a narrow set of tactics—particularly the sum-of-squares approach using `nlinarith`—indicating limited diversity in their proof strategies when it comes to inequalities (Section 3).

4. We find that simple fine-tuning on synthetic compositional problems fails to generalize.** While performance improves on problems similar to the training data, the skill does not transfer to new seed problems, suggesting that data augmentation alone is insufficient to teach robust compositional reasoning.

## 2 Dataset Curation

We introduce Ineq-Comp, a benchmark designed to evaluate the compositional reasoning ability of formal theorem provers in the context of mathematical inequalities. The benchmark is built upon a curated set of seed problems—introductory-level inequality problems solvable using standard techniques such as AM-GM, Cauchy-Schwarz, or Jensen's inequality. Each seed problem is formally proven in Lean 4, ensuring correctness and providing high-quality human-written proofs. From these seed problems, Ineq-Comp is extended into three components:

1. Ineq-Simp (Section 2.1): 150 problems created by applying human-intuitive transformations to the 75 seed problems. These include algebraic rewrites and compositional constructions (e.g., duplicating variables or multiplying inequalities), as illustrated in Figure 1 (Left). Problems are categorized by the technique used in the seed (AM-GM, Cauchy, or Miscellaneous).

2. Ineq-Mix (Section 2.2): A rule-based, automated framework that generates new composed inequalities from any given seed set. This enables scalable extension of Ineq-Simp while preserving formal correctness.

3. Ineq-Real (Section 2.3): A set of 50 real-world inequality problems collected from math contests and educational resources to enhance benchmark diversity and assess generalization beyond synthetic construction.

## 2.1  Ineq-Simp: Simple transformations from seed inequalities

**Seed problems**  We curate 75 seed problems based on direct applications of classical inequalities commonly used in math Olympiads. These include 25 problems using the AM-GM inequality, 25 using Cauchy–Schwarz, and 25 others covering Jensen's inequality (10), Schur's inequality (5), the sum-of-squares method (5), and induction (5). We emphasize AM-GM and Cauchy due to their foundational role in inequality reasoning. Each problem is accompanied by a verified Lean 4 proof, contributing to formal proof resources and ensuring correctness.

**Type I variant**  We generate the type I variant from the seed problem by duplicating the original inequality using distinct variable names and combine the two resulting inequalities. More precisely, given a seed inequality defined on variables $X = (x_1, \ldots, x_n)$, with condition $X \in \mathcal{C}$ and statement $f(X) \geq g(X)$. We replicate the problem with new variables $Y = (y_1, \ldots, y_n)$ under identical conditions, yielding a second inequality: $f(Y) \geq g(Y)$, and we combine the two as follows: (1) If $g(X) \geq 0$, then we multiply two of them together, resulting in a new problem with variables $(X, Y)$, with condition $(X, Y) \in \mathcal{C} \times \mathcal{C}$ and statement $f(X) \cdot f(Y) \geq g(X) \cdot g(Y)$; (2) If $g(X)$ is not guaranteed to be non-negative, we add the inequalities: $f(X) + f(Y) \geq g(X) + g(Y)$. These variants remain structurally identical to seed problems, and humans typically solve them by decomposing into two identical parts. Thus, Type I serves as a minimal test of compositional reasoning: we expect any model that can solve the seed problem to solve the variant with minimal additional reasoning.

**Type II variant**  Type II problems apply a one-step algebraic transformation to the variables of the seed problem. For example, each variable may be squared or square-rooted. Mathematically, given any inequality problem defined on variables $X = (x_1, \ldots, x_n)$, with condition $X \in \mathcal{C}$ and statement $f(X) \geq g(X)$, we define a transformation $T : \mathbb{R}^n \to \mathbb{R}^n$, and we would like to show that with the condition $T(X) \in \mathcal{C}$, $f(T(X)) \geq g(T(X))$. Type II problems test "low-level" compositionality. These problems retain the structure of the seed, differing only by small and systematic substitutions, and are typically easy for humans once the transformation is recognized.

**Quality control**  All seed problems and Lean 4 proofs are curated by individuals with national or international math Olympiad experience. The problems are drawn from fundamental techniques such as AM-GM, Cauchy–Schwarz, Jensen, and Schur, and are kept at or below introductory Olympiad difficulty. Many can be solved by a single application of a well-known inequality, making them accessible to LLMs' capability of informal mathematical reasoning.

## 2.2  Ineq-Mix: A fully expandable benchmark for composition

We introduce Ineq-Mix, an automated framework that generates a potentially unbounded set of valid inequality problems by applying predefined transformation rules to seed problems. These rules fall into three categories: (1) **Compositional transformations** (e.g., combining inequalities via addition, multiplication, max/min); (2) **Variable-level transformations** (e.g., replacing variables with algebraic expressions); (3) **Problem-level transformations** (e.g., applying a monotonic function like $\exp$ or $\log$ to both sides). All generated problems are guaranteed to be mathematically valid and formally solvable. Unlike the simpler variants in Section 2.1, Ineq-Mix produces more complex compositions that often require nontrivial decomposition and planning. For example, proving $f_1 f_2 \geq g_1 g_2$ from $f_1 \geq g_1$ and $f_2 \geq g_2$ may require identifying intermediate inequalities. We generate 100 evaluation problems using only compositional rules (Table 4), which already challenge state-of-the-art provers. As models improve, Ineq-Mix can scale in complexity by combining multiple transformation types.

To assess whether this difficulty is inherent to formal reasoning, we test OpenAI's O3 model, which operates in natural language, on problems involving one transformation from each category. O3 reliably decomposes and solves these using intuitive reasoning, with failures typically due to mistakes on individual subproblems. This contrast highlights a key gap: informal models like O3 handle compositionality with ease, while formal provers struggle even on problems built from elementary components. This suggests the bottleneck lies in current formal reasoning systems, not the mathematical content. See Appendix B for a full list of transformations used in Ineq-Mix.

### 2.3 Ineq-Real: real-world inequalities

The Ineq-Real subset includes 50 inequality problems sourced from real-world materials, such as math competitions, training materials, and Chinese Gaokao exams. Each problem is manually translated into Lean 4 and verified by individuals with IMO/CMO-level expertise.

Although the core techniques—such as AM-GM, Cauchy-Schwarz, and Schur—overlap with those in Ineq-AMGM, Ineq-Cauchy, and Ineq-MISC, the Ineq-Real problems are generally more complex. They often require combining multiple techniques, deeper insight, or nontrivial algebraic manipulation, rather than straightforward application of a single inequality. Unlike the modular problems in earlier subsets, Ineq-Real tests the application of known tools in real-world contexts and naturally complements the synthetic, semi-structured problems in Ineq-Comp.

## 3 Experiments and Findings

### 3.1 Experiment setup

**Models and methods.** We evaluate three categories of models and methods: (1) General-purpose language models, (2) Whole-proof generation models, and (3) Tree-search methods. For general-purpose models, we test DeepSeek-R1-Distill-Qwen-32B (Guo et al., 2025) in two settings: with and without explicit thinking steps. Without the chat-style prompt, the model directly completes the Lean 4 proof code. In contrast, when provided with a chat template and generation prompt, the model first performs a "thinking process" in natural language before outputting the full Lean script. For whole-proof generation, we include DeepSeek-Prover-V1.5-RL (Xin et al., 2024), Goedel-Prover-SFT (Lin et al., 2025a), STP (Dong and Ma, 2025), Kimina-7B (Wang et al., 2025), and the latest state-of-the-art model DeepSeek-Prover-V2-7B (Ren et al., 2025), which achieves top performance on the MiniF2F benchmark. For tree-search-based approaches, we evaluate DeepSeek-Prover-V1.5-RL+RMaxTS (Xin et al., 2024) and InternLM-2.5-Step-Prover+BF (Wu et al., 2024). While recent models such as HunyuanProver (Li et al., 2024) and BFS-Prover (Xin et al., 2025) show stronger performance on MiniF2F, we exclude them due to the lack of publicly available code.

**Budget and evaluation metric.** We evaluate all methods using the standard pass@$N$ accuracy, where a problem is considered solved if at least one of $N$ generated proofs is correct. For tree-search methods, budget denotes the total number of interactions with the Lean compiler. For additional details, see Appendix D, and refer to Appendix E for the prompt templates used in evaluation.

### 3.2 Main findings and discussion

The experiment results on Ineq-Simp: categorized by the natural of seed problems Ineq-AMGM, Ineq-Cauchy, and Ineq-MISC are summarized in Table 1. The performance of different models on Ineq-Mix (randomly generated) and Ineq-Real is shown in Figure 2.

**Lack of compositional ability.** Our experiments reveal a clear failure of compositional generalization across most current theorem provers, even on problems derived from simple seed inequalities that remain easy for humans. As shown in Table 1, top models such as Goedel-Prover-SFT (Lin et al., 2025a), STP (Dong and Ma, 2025), and Kimina-Prover (Wang et al., 2025) show large drops in accuracy from seed problems to their compositional variants: Type I (duplication-based) and Type II (algebraic transformation).

For instance, Kimina-Prover-Preview-Distill-7B achieves 80% on AM-GM seed problems (budget 3200), but drops to 44% on Type I and 64% on Type II. The drop is even more dramatic for Goedel-Prover and DeepSeek-Prover-V1.5, which struggle to exceed 5% accuracy on Ineq-AMGM Type I problems—even with a large number of attempts. The only partial exception is DeepSeek-Prover-V2-7B, which reduces the seed-to-Type I gap to 13% and seed-to-Type II to 18%. Its drop becomes more pronounced at smaller budgets (e.g., pass@32), confirming that the challenge is not just probabilistic but fundamentally hard (Figure 1, right). This is especially striking given that DeepSeek-Prover-V2 is explicitly trained using a divide-and-conquer strategy (Ren et al., 2025), which should, in principle, help with decomposing Type I problems. Its performance gap, particularly under low-pass settings, highlights the intrinsic difficulty of compositional reasoning in formal reasoning.

Table 1: Performance of different models and methods on Ineq-Simp (Ineq-AMGM, Ineq-Cauchy, and Ineq-MISC). We report the pass@$N$ accuracy and its standard deviation (subscript text).

| Method | Budget | Ineq-AMGM | | | Ineq-Cauchy | | | Ineq-MISC | | |
|---|---|---|---|---|---|---|---|---|---|---|
| | | Seed | Type I | Type II | Seed | Type I | Type II | Seed | Type I | Type II |
| *General Purpose Models* | | | | | | | | | | |
| DeepSeek-R1-Distill-Qwen-32B (w/o thinking)(Guo et al., 2025) | 32 | $48.2_{1.9}$ | $3.5_{3.3}$ | $16.2_{3.0}$ | $28.0_{3.3}$ | $17.0_{3.2}$ | $15.0_{3.0}$ | $41.4_{3.7}$ | $13.4_{4.5}$ | $15.4_{4.4}$ |
| | 64 | $49.0_{1.7}$ | $6.5_{4.1}$ | $18.4_{2.4}$ | $30.6_{3.2}$ | $19.5_{2.8}$ | $16.8_{2.7}$ | $44.5_{3.2}$ | $17.7_{4.0}$ | $20.2_{4.8}$ |
| | 128 | $49.9_{2.0}$ | $10.6_{4.2}$ | $20.0_{2.5}$ | $32.6_{2.9}$ | $21.8_{3.2}$ | $19.0_{2.6}$ | $47.4_{3.1}$ | $21.1_{3.7}$ | $25.4_{4.2}$ |
| | 3200 | 52.0 | 40.0 | 36.0 | 44.0 | 32.0 | 28.0 | 52.0 | 36.0 | 36.0 |
| DeepSeek-R1-Distill-Qwen-32B (w thinking)(Guo et al., 2025) | 32 | $48.8_{1.6}$ | $10.9_{3.8}$ | $21.1_{3.1}$ | $42.9_{2.5}$ | $27.0_{3.4}$ | $18.4_{2.4}$ | $50.5_{2.3}$ | $18.9_{4.6}$ | $22.0_{4.0}$ |
| | 64 | $49.5_{1.9}$ | $14.5_{4.4}$ | $23.0_{3.4}$ | $44.5_{2.4}$ | $30.3_{2.9}$ | $20.6_{2.3}$ | $51.9_{0.6}$ | $23.7_{4.9}$ | $26.2_{3.1}$ |
| | 128 | $50.9_{2.1}$ | $19.2_{4.1}$ | $26.1_{4.3}$ | $46.2_{2.3}$ | $32.6_{2.7}$ | $22.1_{2.0}$ | $52.0_{0.0}$ | $28.0_{3.9}$ | $29.4_{2.7}$ |
| | 3200 | 60.0 | 44.0 | 44.0 | 56.0 | 40.0 | 24.0 | 52.0 | 36.0 | 40.0 |
| *Whole-Proof Generation Methods* | | | | | | | | | | |
| DeepSeek-Prover-V1.5-RL (Xin et al., 2024) | 32 | $48.1_{3.0}$ | $0.0_{0.4}$ | $8.2_{1.5}$ | $14.9_{3.2}$ | $2.9_{1.8}$ | $4.4_{1.4}$ | $40.2_{2.8}$ | $12.4_{1.1}$ | $12.2_{2.5}$ |
| | 64 | $50.6_{2.9}$ | $0.1_{0.6}$ | $9.0_{1.7}$ | $17.0_{2.7}$ | $3.7_{1.1}$ | $5.0_{1.9}$ | $42.1_{2.3}$ | $12.7_{1.7}$ | $13.8_{2.9}$ |
| | 128 | $52.2_{2.1}$ | $0.2_{0.8}$ | $9.8_{2.0}$ | $18.7_{2.7}$ | $4.0_{0.0}$ | $6.1_{2.3}$ | $43.2_{1.6}$ | $13.3_{2.2}$ | $16.2_{2.9}$ |
| | 3200 | 60.0 | 4.0 | 24.0 | 24.0 | 4.0 | 12.0 | 44.0 | 20.0 | 28.0 |
| Goedel-Prover-SFT (Lin et al., 2025a) | 32 | $48.6_{2.9}$ | $0.4_{1.2}$ | $14.0_{3.2}$ | $34.8_{2.5}$ | $12.4_{3.5}$ | $21.5_{3.4}$ | $47.0_{1.7}$ | $14.4_{3.1}$ | $24.6_{1.9}$ |
| | 64 | $50.6_{2.6}$ | $0.8_{1.6}$ | $16.6_{2.8}$ | $36.2_{1.9}$ | $15.8_{3.4}$ | $24.6_{2.9}$ | $47.8_{0.9}$ | $16.6_{2.5}$ | $25.5_{1.9}$ |
| | 128 | $52.2_{1.4}$ | $1.3_{1.9}$ | $18.6_{2.2}$ | $37.1_{1.8}$ | $19.4_{2.9}$ | $26.9_{1.8}$ | $48.0_{0.0}$ | $17.9_{2.6}$ | $26.4_{2.5}$ |
| | 3200 | 60.0 | 4.0 | 24.0 | 40.0 | 32.0 | 28.0 | 48.0 | 24.0 | 36.0 |
| STP (w/o miniF2F valid) (Dong and Ma, 2025) | 32 | $59.1_{1.9}$ | $14.3_{4.4}$ | $23.2_{4.5}$ | $35.2_{2.4}$ | $14.6_{2.7}$ | $16.0_{2.6}$ | $55.6_{1.3}$ | $12.6_{5.0}$ | $27.6_{3.6}$ |
| | 64 | $60.1_{0.6}$ | $18.5_{4.1}$ | $28.2_{4.6}$ | $36.8_{2.4}$ | $16.7_{2.8}$ | $17.3_{2.7}$ | $56.0_{0.0}$ | $17.8_{4.9}$ | $31.0_{4.1}$ |
| | 128 | $60.3_{1.1}$ | $24.3_{4.1}$ | $33.0_{3.6}$ | $37.9_{2.6}$ | $18.4_{3.0}$ | $18.9_{3.3}$ | $56.0_{0.0}$ | $24.0_{4.4}$ | $33.9_{4.1}$ |
| | 3200 | 64.0 | 44.0 | 40.0 | 44.0 | 24.0 | 28.0 | 56.0 | 36.0 | 40.0 |
| Kimina-Prover-Preview-Distill-7B (Wang et al., 2025) | 32 | $59.4_{4.1}$ | $11.7_{5.4}$ | $45.2_{3.7}$ | $46.9_{4.5}$ | $27.0_{2.6}$ | $27.7_{3.3}$ | $44.2_{1.3}$ | $18.1_{3.9}$ | $35.8_{2.0}$ |
| | 64 | $64.1_{4.6}$ | $19.4_{5.9}$ | $48.6_{2.4}$ | $52.7_{4.3}$ | $28.8_{2.5}$ | $30.2_{2.8}$ | $44.6_{1.4}$ | $22.3_{2.9}$ | $36.8_{2.0}$ |
| | 128 | $69.4_{4.2}$ | $28.2_{5.4}$ | $50.6_{2.2}$ | $57.6_{3.6}$ | $30.4_{3.0}$ | $32.0_{1.6}$ | $45.1_{1.8}$ | $25.6_{2.5}$ | $37.6_{2.5}$ |
| | 3200 | 80.0 | 44.0 | 64.0 | 68.0 | 52.0 | 36.0 | 52.0 | 32.0 | 44.0 |
| DeepSeek-Prover-V2-7B (Ren et al., 2025) | 32 | $75.0_{4.4}$ | $58.6_{4.0}$ | $52.5_{4.5}$ | $64.6_{4.1}$ | $33.0_{2.3}$ | $35.0_{2.3}$ | $59.1_{2.9}$ | $49.3_{3.4}$ | $38.8_{4.4}$ |
| | 64 | $80.7_{5.3}$ | $62.1_{4.5}$ | $57.4_{4.0}$ | $68.3_{3.1}$ | $34.7_{2.7}$ | $36.6_{2.3}$ | $61.7_{2.5}$ | $51.6_{2.9}$ | $43.7_{4.2}$ |
| | 128 | $85.8_{5.4}$ | $65.9_{5.3}$ | $61.4_{3.7}$ | $71.0_{2.0}$ | $36.3_{3.6}$ | $37.9_{2.6}$ | $64.0_{1.6}$ | $53.3_{3.1}$ | $49.9_{4.3}$ |
| | 3200 | 96.0 | 84.0 | 76.0 | 76.0 | 52.0 | 48.0 | 68.0 | 64.0 | 64.0 |
| *Tree Search Methods* | | | | | | | | | | |
| DeepSeek-Prover-V1.5-RL +RMaxTS(Xin et al., 2024) | 1×3200 | $60.0_{0.0}$ | $3.0_{1.7}$ | $22.0_{2.0}$ | $24.0_{0.0}$ | $8.0_{2.8}$ | $13.0_{3.3}$ | $44.0_{0.0}$ | $14.0_{3.5}$ | $29.0_{1.7}$ |
| | 2×3200 | $60.0_{0.0}$ | $6.0_{2.0}$ | $26.0_{2.0}$ | $24.0_{0.0}$ | $10.0_{2.0}$ | $16.0_{0.0}$ | $44.0_{0.0}$ | $16.0_{4.0}$ | $32.0_{0.0}$ |
| | 4×3200 | 60.0 | 8.0 | 28.0 | 24.0 | 12.0 | 20.0 | 44.0 | 20.0 | 36.0 |
| InternLM2.5-StepProver+BF (Wu et al., 2024) | 1×32×600 | $30.8_{3.1}$ | $0.0_{0.0}$ | $2.5_{3.1}$ | $12.0_{0.0}$ | $0.0_{0.0}$ | $1.2_{1.9}$ | $34.0_{2.0}$ | $2.2_{2.0}$ | $17.0_{3.9}$ |
| | 4×32×600 | $38.0_{4.5}$ | $0.0_{0.0}$ | $9.0_{3.3}$ | $12.0_{0.0}$ | $0.0_{0.0}$ | $3.0_{1.7}$ | $36.0_{0.0}$ | $5.0_{1.7}$ | $21.0_{1.7}$ |
| | 16×32×600 | 44.0 | 0.0 | 24.0 | 12.0 | 0.0 | 4.0 | 36.0 | 8.0 | 24.0 |

Besides, tree-search methods can not easily bridge the compositional gap: both DeepSeek-Prover-V1.5-RL+RMaxTS and InternLM-2.5-StepProver+BF struggle with even the simple Type I and Type II variants, showing that search alone does not resolve the underlying generalization challenge.

When we evaluate more complex compositions using Ineq-Mix, where two seed problems are combined using compositional rules, the difficulty increases further (Figure 2, left). Out of 100 such problems, all whole-proof generation models except DeepSeek-Prover-V2-7B solve fewer than 8, while DeepSeek-Prover-V2-7B significantly leads with 22 solves under a 128-trial budget.

These results underscore a key limitation of most LLM-based theorem provers: while they can handle familiar or isolated problems, they fail to generalize reasoning strategies across even modest compositions. Their inability to reuse known proofs in structured combinations highlights a major bottleneck in formal mathematical reasoning.

**Reliance on sum-of-squares and lack of high-level knowledge.** A second key finding is that LLM-based theorem provers overwhelmingly rely on low-level algebraic tactics and rarely apply classical inequalities such as AM-GM, Cauchy-Schwarz, or Jensen's inequality, even when those are the most appropriate tools. Despite the benchmark being designed to encourage such techniques, no model except Kimina-Distill-7B apply them directly. Most models default to Lean's `nlinarith` tactic with `sq_nonneg`, reducing problems to sum-of-squares.

Interestingly, current provers that generate informal reasoning, such as natural language comments or the thinking drafts, often do produce correct high-level strategies in natural language, frequently mentioning the appropriate inequality (e.g., AM-GM) or correctly describe how to decompose a

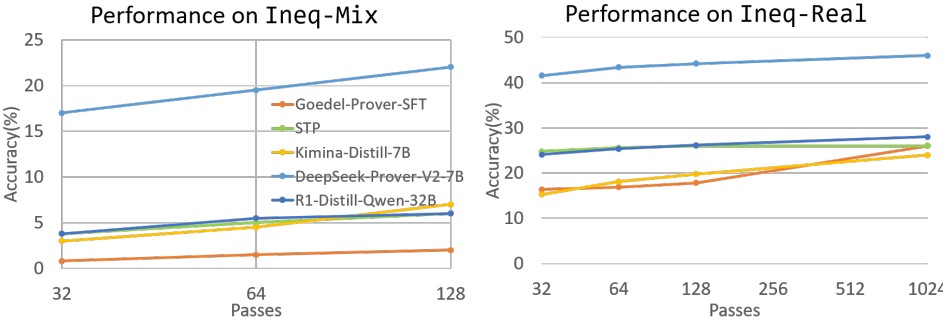

Figure 2: Pass@N accuracy of various LLM-based theorem provers on the Ineq-Mix (Left) and Ineq-Real (Right) subsets under increasing computational budgets (x-axis). Surprisingly, the automatically generated Ineq-Mix problems—created by applying simple, human-intuitive compositional transformations to pairs of seed inequalities—are more challenging for current models than the real-world problems in Ineq-Real. Most models, including Goedel-Prover, STP, and Kimina, solve fewer than 7% of Ineq-Mix problems even with 128 attempts, while DeepSeek-Prover-V2 achieves a modest 22%. In contrast, on Ineq-Real, the model performance are at least doubled. This result highlights a critical weakness in formal theorem provers: reasoning compositionally across structurally simple subproblems is harder than solving complex, real-world inequalities.

problem into parts. However, many of these strategies are abandoned in the corresponding Lean code, and models apply brute-force algebraic manipulation instead of high-level knowledge.

This gap underscores a major limitation: while models may recognize the right approach in natural language, they struggle to implement it in formal proof steps. See Appendix F for examples.

**A different viewpoint from existing benchmarks.** Ineq-Comp offers a complementary lens to existing benchmarks like MiniF2F. While strong models such as DeepSeek-Prover-V2-7B and Kimina-Distill-7B perform well on both, discrepancies emerge that reveal what each benchmark truly measures. For example, Goedel-Prover-SFT (Lin et al., 2025a) slightly outperforms STP (without MiniF2F validation augmentation) (Dong and Ma, 2025) on MiniF2F, but is surpassed by STP on Ineq-Comp, which targets compositional generalization. Similarly, InternLM2.5-StepProver+BF (Wu et al., 2024) performs comparably to both on MiniF2F, but struggles significantly on Ineq-Comp, especially on Type I problems, revealing weak high-level compositional reasoning.

These contrasts highlight Ineq-Comp 's focus on generalization over structured transformations, making it a valuable complement to broader benchmarks like MiniF2F.

# 4 Probing and Mitigating Compositional Challenges

To better understand and potentially mitigate the compositional failures observed in Section 3, we conduct two ablation studies. First, we explore whether our curated benchmark becomes easier when models' sizes are scaled up (Section 4.1), which tests if the compositional failure is unique to relatively small models and can be easily solved by scaling up. Next, we investigate where our benchmark becomes easier when models are provided with the formal proof of the seed problem as part of the input (Section 4.2). This tests whether models can leverage in-context demonstrations to generalize to the transformed variants. Building on this idea, we then construct synthetic data that explicitly encodes compositional structure and use it to fine-tune the theorem prover. This examines whether the lack of compositional ability can be remedied through supervised fine-tuning alone (Section 4.3).

## 4.1 Scaling-up model parameters

In Section 3, we mostly test theorem provers with sizes at most 32B, and we do not test provers with different sizes within the same family. Thus, it is natural to ask if the compositional weakness pinpointed in Section 3 can be easily solved by scaling up model parameters.

To answer the question, we test DeepSeek-Prover-V2 7B and its 671B counterpart (Ren et al., 2025), as well as Goedel-Prover-V2 8B and its 32B counterpart (Lin et al., 2025b), under pass@32 on

Table 2: Performance of models with different sizes on Ineq-Comp under pass@32.

| Model | Seed | Type I | Type II |
|---|---|---|---|
| DeepSeek-Prover-V2-7B (Ren et al., 2025) | 66.23 | 46.97 | 42.1 |
| DeepSeek-Prover-V2-671B | 88.0 | 68.33 | 70.67 |
| Goedel-Prover-V2-8B (Lin et al., 2025b) | 69.0 | 35.0 | 49.0 |
| Goedel-Prover-V2-32B | 90.0 | 68.0 | 83.0 |

Table 3: **ICL-based ablation.** Performance of various models on Ineq-Simp (Type I and Type II variants) under the in-context learning (ICL) setting. For each test instance, the model is provided with the full formal proof of the corresponding seed problem as part of the input prompt. This setup evaluates whether models can leverage prior solutions to solve transformed variants via compositional generalization. We report pass@$N$ accuracy (with standard deviation as subscript) across different generation budgets $N$. Despite access to a complete solution to the seed problem, all models—including proprietary ones—continue to struggle on both Type I and Type II variants, indicating that the compositional gap cannot be easily closed through in-context demonstrations alone.

| Models | Budget (# Proofs) | Ineq-AMGM | | Ineq-Cauchy | | Ineq-MISC | |
|---|---|---|---|---|---|---|---|
| | | Type I | Type II | Type I | Type II | Type I | Type II |
| *Open-Source Models* | | | | | | | |
| Qwen2.5-Coder-32B-Instruct (Hui et al., 2024) | 32 | $11.0_{1.7}$ | $28.0_{4.9}$ | $34.0_{6.0}$ | $9.0_{3.3}$ | $39.0_{3.3}$ | $44.0_{7.5}$ |
| | 64 | $22.0_{2.0}$ | $34.0_{2.0}$ | $40.0_{8.0}$ | $14.0_{6.0}$ | $46.0_{2.0}$ | $52.0_{4.0}$ |
| | 128 | 28.0 | 40.0 | 48.0 | 20.0 | 56.0 | 60.0 |
| DeepSeek-R1-Distill-Qwen-32B (w/o thinking)(Guo et al., 2025) | 32 | $6.0_{4.5}$ | $43.0_{3.3}$ | $52.0_{4.9}$ | $27.0_{4.4}$ | $31.0_{3.3}$ | $39.0_{5.9}$ |
| | 64 | $10.0_{2.0}$ | $50.0_{2.0}$ | $68.0_{4.0}$ | $32.0_{0.0}$ | $48.0_{0.0}$ | $46.0_{2.0}$ |
| | 128 | 16.0 | 56.0 | 76.0 | 44.0 | 64.0 | 56.0 |
| DeepSeek-R1-Distill-Qwen-32B (w thinking)(Guo et al., 2025) | 32 | $8.0_{4.9}$ | $39.0_{1.7}$ | $64.0_{2.8}$ | $28.0_{6.3}$ | $35.0_{7.7}$ | $45.0_{4.4}$ |
| | 64 | $12.0_{0.0}$ | $42.0_{2.0}$ | $78.0_{2.0}$ | $34.0_{2.0}$ | $44.0_{8.0}$ | $52.0_{0.0}$ |
| | 128 | 16.0 | 44.0 | 84.0 | 40.0 | 56.0 | 60.0 |
| *Proprietary Models* | | | | | | | |
| GPT-4o | 16 | 12.0 | 40.0 | 56.0 | 16.0 | 44.0 | 60.0 |
| Claude 3.7 Sonnet (w/o thinking) | 16 | 36.0 | 28.0 | 32.0 | 20.0 | 32.0 | 24.0 |

Ineq-Comp. Table 2 summarizes the result, which shows that although scaling up the model size improves the absolute performance, the relative gap between the Type I, II problems and the seed problems still exists, implying that the compositional weakness in formal theorem proving is not easily solvable by brute-force scaling up model parameters.

### 4.2 In-context learning

To understand whether access to the original proof helps models solve transformed inequality problems, we conduct an in-context learning (ICL) ablation. For each Type I and Type II problem, models are provided with the complete, verified Lean 4 proof of the corresponding seed problem in the prompt. This setup tests whether models can leverage known solutions to generalize compositionally.

Table 3 reports the pass@$N$ accuracy across a range of open-source and proprietary models on Ineq-AMGM, Ineq-Cauchy, and Ineq-MISC problems under different generation budgets. Despite having access to the seed solution, all models continue to struggle with the transformed variants. For example, Qwen2.5-Coder-32B-Instruct achieves only 40.0% accuracy on Type II AM-GM problems and 20.0% on Type II Cauchy–Schwarz problems at 128 attempts. DeepSeek-R1-Distill-Qwen-32B (with thinking) performs similarly, reaching just 16.0% on Type I AM-GM problems. Proprietary models such as GPT-4o and Claude 3.7 Sonnet show mixed results, further illustrating the challenge. These findings suggest that simply providing the seed proof is not enough to enable compositional generalization. Even modest transformations remain difficult, pointing to a fundamental limitation in their ability to transfer and reuse formal reasoning strategies.

### 4.3 Fine-tuning

Previous studies, such as Zhao et al. (2024); Abedsoltan et al. (2025), have demonstrated that simple skill composition can be effectively learned through naive fine-tuning, showing generalization even in out-of-distribution (OOD) scenarios. Motivated by this, we conducted a fine-tuning experiment to

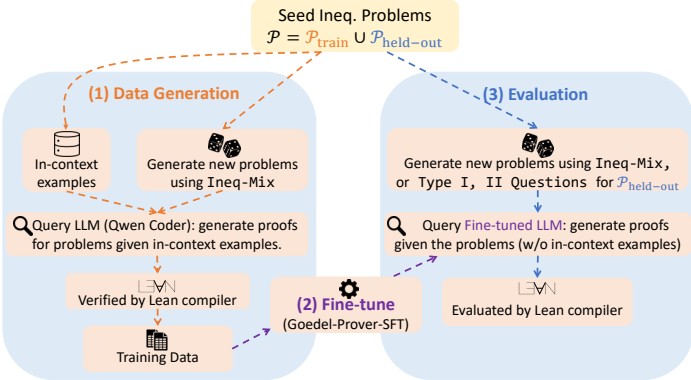

Figure 3: Pipeline for evaluating if composition can be learned through naive fine-tuning. We split the seed problems into $\mathcal{P}_{\text{train}}$, which contains 25 problems utilizing AMGM, and $\mathcal{P}_{\text{held-out}}$, which contains the remaining 50 problems. The pipeline consists of three steps: (1) generate data by prompting LLM given the proof of seed problems in context and keep only the proofs that pass Lean compilation; (2) fine-tune Goedel-Prover-SFT; (3) evaluate type I and II problems and Ineq-Mix on $\mathcal{P}_{\text{held-out}}$.

explore whether our composition and algebraic transformation structures could similarly enhance model performance in formal inequality proving.

We split the seed problems into two subsets: 25 AM-GM inequality seed problems as training set and 50 other problems as held-out set. To construct training data, we use our automated framework (Ineq-Mix) to generated 15k problems incorporating both algebraic and compositional transformations base on the AM-GM seed problems. Then we used in-context learning (ICL) to prompt Qwen2.5-Coder-32B-Instruct with seed problems and their verified Lean 4 proofs. We retained approximately 8k compilable outputs and fine-tuned Goedel-Prover-SFT on this data. See Figure 3 for an overview of the process and Appendix D for setup details.

Fine-tuning led to substantial in-distribution (ID) improvements, especially on Type I AM-GM problems, where accuracy rose to 56.0% at a high generation budget. However, gains on out-of-distribution (OOD) tasks—such as Type II or unseen algebraic transformations—were minimal. Even with 3200 attempts, performance on these variants remained close to pre-finetuning levels.

These findings suggest that while models can learn to replicate specific compositional structures from data, their ability to generalize this reasoning to structurally similar but unseen cases remains limited. This highlights that compositional reasoning in formal proofs may not be easily acquired through moderate-scale supervised fine-tuning alone. Addressing this challenge likely requires significantly richer training signals (Ren et al., 2025). Notably, the fine-tuned model retains its original performance on MiniF2F, confirming that the lack of OOD gains is not due to catastrophic forgetting, but stems from fundamental limitations in current prover capabilities.

## 5 Related Works

**Automated theorem proving (ATP)** ATP has been a long-standing goal in AI (Wu, 2008). Modern theorem provers use tactic-based proof search and premise selection within these systems (e.g. Lean's tactics in Mathlib environment), but they struggle with enormous search spaces and limited training data for complex theorems (Polu and Sutskever, 2020; Yang et al., 2023). More recently, more theorem provers came out based on large language models (Xin et al., 2024; Lin et al., 2025a; Dong and Ma, 2025; Wang et al., 2025; Ren et al., 2025), and also possibly incorporated tree-search techniques (Wu et al., 2024; Li et al., 2024; Xin et al., 2025).

While achieving better performance on standard benchmarks like MiniF2F, our Ineq-Comp benchmark shows that these models struggle with even simple composition. There are also inequality solvers targeting much harder math Olympiad-level inequalities that best LLMs like GPT-4 cannot solve (Wei et al.; Li et al., b). These methods are not taken into consideration since our benchmark

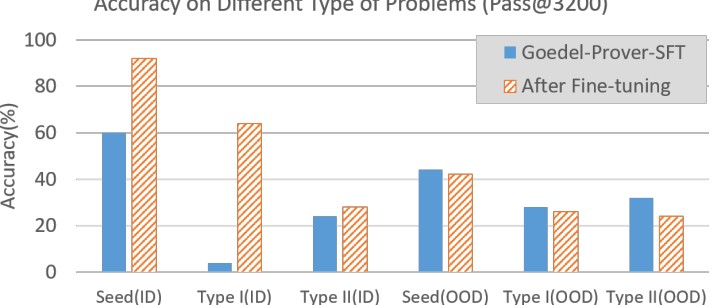

Figure 4: **SFT base ablation.** The performance of the fine-tuned model on different type of problems in Ineq-AMGM (ID), Ineq-Cauchy (OOD) and Ineq-MISC (OOD). We group different types of problems in Ineq-Cauchy and Ineq-MISC together as OOD and report the accuracy under Pass@3200. Although fine-tuning significantly improved in-distribution (ID) Type I performance, it did not lead to meaningful improvements on out-of-distribution (OOD) generalization tasks or algebraic transformation problems.

focuses on general theorem provers and shows that they cannot solve the math problems trivial to human and can be easily solved by LLMs through natural language reasoning.

**Datasets and benchmarks for ATP.** Formal libraries like Lean's such as `mathlib` (Lean) (van Doorn et al., 2020), the Archive of Formal Proofs (Isabelle), and the Coq Standard Library provide large collections of high-quality formal proofs. However, these proofs tend to reflect routine reasoning with limited diversity in problem structure. Datasets extracted from these libraries, such as CoqGym (Yang and Deng, 2019) and IsarStep (Li et al., a), provide goal–tactic supervision, but they fall short in capturing the kind of creative or compositional reasoning in math competitions. To assess generalization, several curated benchmarks have been proposed. MiniF2F (Zheng et al., 2021) aggregates problems from math contests ranging from AMC to IMO, spanning a broad difficulty range but lacking fine-grained structure, often mixing trivial exercises with advanced Olympiad problems. ProofNet (Azerbayev et al., 2023) focuses on undergraduate-level mathematics, while PutnamBench (Tsoukalas et al., 2024) targets advanced problems from the Putnam competition, which remain unsolved by most current models.

AIPS (Wei et al.) and Li et al. (b) introduce (synthetic) datasets for inequality proving, targeting IMO-level challenges using symbolic–neural methods. These efforts aim to push the boundary of inequality proving abilities. While these efforts highlight the complexity and elegance of inequality proving, they primarily aim to push the limits of problem difficulty. In contrast, Ineq-Comp takes a bottom-up approach: starting from simple, human-solvable inequalities and applying minimal, structured transformations to test compositional generalization. Rather than measuring raw difficulty, we isolate a model's ability to reuse reasoning across structurally varied but logically equivalent problems. While AIPS probes advanced inequality solving, Ineq-Comp reveals brittleness on easy problems, exposing a different and underexplored axis of failure.

## 6 Conclusion

We introduced a benchmark to evaluate compositional generalization in automated theorem proving, centered on inequality problems. By applying simple, human-intuitive transformations to seed problems, we revealed significant performance drops across LLM-based provers, highlighting a core weakness in compositional reasoning. These findings point to an important frontier for formal reasoning systems: developing models that not only learn individual proof strategies, but can reliably compose and adapt them in a formally verifiable manner.

## Acknowledgement

The authors would like to thank Danqi Chen and Kaiyu Yang for helpful discussions. HZ and SA acknowledge support from NSF, ONR, and Schmidt Foundation. CJ acknowledge the support from NSF-OAC-2411299 and NSF-IIS-2239297.

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

# Appendix

## A  Limitation

The main limitation of our work is the amount of computational resources required, which prevented us from conducting an extensive hyperparameter search across all models. For instance, the generation temperature was fixed at 1.0 under all generation budgets, and the maximum generation length was set to be 16k. Our experiments cost over 10,000 H100 GPU hours in total. Despite the lack of an extensive hyperparameter search, the observed performance differences in our results are significantly larger than standard deviation. We therefore believe that our main conclusions would hold even if more comprehensive hyperparameter tuning were conducted.

## B  More Details on Ineq-Mix

In this section, we present the details of Ineq-Mix, including the technical requirements and all the transformation rules for different categories.

**Seed problems**  We first focus on the seed problems. Every seed problem takes the following form:

$$
\begin{aligned}
\text{Variables:} \quad & X = \{x_1, \ldots, x_n\} \in \mathbb{R}^n \\
\text{Basic assumptions:} \quad & X \succ 0 \quad (x_i > 0, \forall i \in [n]) \\
\text{Further assumptions:} \quad & X \in \mathcal{C} \\
\text{Statement:} \quad & f(x_1, \ldots, x_n) \geq g(x_1, \ldots, x_n) \\
\text{Positive RHS:} \quad & \text{if rhs } g(x_1, \ldots, x_n) \text{ is guaranteed to be positive.}
\end{aligned}
$$

Here, the variables need to be a set with fixed number of variables, i.e., the number of variables need to be constant. For inequalities with infinite number of variables or with $n$-variables, they are not taken into consideration as seed problems. We also assume that all the variables are real numbers.

We also add the assumption for every seed problem that $x_i > 0$ for all $i \in [n]$. This assumption ensures the correctness for composition and further algebraic transformations.

The seed inequality problem can have further conditions on the variables. For example, for a 3-variable inequality with variables $a, b, c$, a possible additional assumption is $a + b + c = 1$.

Finally, we also have a boolean tag to denote if the right-hand side of the statement $g(x_1, \ldots, x_n)$ is guaranteed to be positive, under the assumption that $X \succ 0$ and $X \in \mathcal{C}$. This tag helps filter out some composition / algebraic transformation rules to ensure the correctness of the transformed problem.

The following are two examples (famous inequalities) that violate our requirement.

*Example 1:* Variant of Bernoulli inequality. For any $x > 0$ and any $n \in \mathbb{N}$, we have

$$(1 + x)^n \geq 1 + nx.$$

This inequality does not satisfy our requirement since $n$ is an integer taken as an input. If we change $n$ to a fixed constant, say 10, then it is a valid seed problem.

*Example 2:* Variant of $n$-variable Cauchy's inequality. For any $n \in \mathbb{N}$ and variables $a_1, a_2, \ldots, a_n$, $b_1, b_2, \ldots, b_n$ with $a_i \in \mathbb{R}^+, b_i \in \mathbb{R}^+, \forall i \in [n]$, we have

$$\left( \sum_{i=1}^{n} a_i^2 \right) \left( \sum_{i=1}^{n} b_i^2 \right) \geq \left( \sum_{i=1}^{n} a_i b_i \right)^2.$$

This inequality does not satisfy our requirement since $n$ is an integer taken as input, and the number of variables is not a fixed contact (it changes with $n$). If we change $n$ to a fixed constant, say 10, then it is a valid seed problem.

Among 75 seed problems from Ineq-AMGM, Ineq-Cauchy, and Ineq-MISC, we get 65 seed problems that satisfies the requirement after small modifications. The discarded problems include some problems that require induction (thus involving integers) or $n$-variable Cauchy's inequality, or some problems that cannot easily add the required assumption $x_i > 0$ for all $i$.

Table 4: **(A list of composition methods for two inequality problems.)** Given two inequality problems $P_1, P_2$, where $P_1$ has variables $X = \{x_1, \ldots, x_m\}$, with the statement: given condition on the variables $C_1(X)$, the following inequality holds $f_1(X) \geq g_1(X)$, and $P_2$ has variables $X$, with the statement: given condition on the variables $C_2(X)$, the following inequality holds $f_2(X) \geq g_2(X)$, the following list defines several composition operations to generate a new inequality problem. If problems $P_1$ and $P_2$ have different set of variables, we can always lift the problems and make sure they are defined on the same variable space. Following are the composition methods we considered to combine $P_1$ and $P_2$ together. If $g_1(X)$ and $g_2(X)$ are both greater than 0, there are more composition methods. We also make sure that $C_1(X) \cap C_2(X)$ is non-empty after composition.

| Name | Statement of the new problem |
|------|------------------------------|
| *Without further condition on $g_1(X)$ and $g_2(X)$.* | |
| Direct Addition | Given condition $C_1(X)$ and $C_2(X)$, show that $f_1(X) + f_2(X) \geq g_1(X) + g_2(X)$. |
| Weighted Sum (for $\mu, \lambda > 0$) | Given condition $C_1(X)$ and $C_2(X)$, show that $\mu f_1(X) + \lambda f_2(X) \geq \mu g_1(X) + \lambda g_2(X)$. |
| Maxima | Given condition $C_1(X)$ and $C_2(X)$, show that $\max\{f_1(X), f_2(X)\} \geq \max\{g_1(X), g_2(X)\}$. |
| Minima | Given condition $C_1(X)$ and $C_2(X)$, show that $\min\{f_1(X), f_2(X)\} \geq \min\{g_1(X), g_2(X)\}$. |
| *With further conditions $g_1(X) > 0$ and $g_2(X) > 0$.* | |
| Direct Addition | Given condition $C_1(X)$ and $C_2(X)$, show that $f_1(X) + f_2(X) \geq g_1(X) + g_2(X) > 0$. |
| Weighted Sum (for $\mu, \lambda > 0$) | Given condition $C_1(X)$ and $C_2(X)$, show that $\mu f_1(X) + \lambda f_2(X) \geq \mu g_1(X) + \lambda g_2(X) > 0$. |
| Maxima | Given condition $C_1(X)$ and $C_2(X)$, show that $\max\{f_1(X), f_2(X)\} \geq \max\{g_1(X), g_2(X)\} > 0$. |
| Minima | Given condition $C_1(X)$ and $C_2(X)$, show that $\min\{f_1(X), f_2(X)\} \geq \min\{g_1(X), g_2(X)\} > 0$. |
| Multiplication | Given condition $C_1(X)$ and $C_2(X)$, show that $f_1(X) \cdot f_2(X) \geq g_1(X) \cdot g_2(X) > 0$. |
| Division | Given condition $C_1(X)$ and $C_2(X)$, show that $\frac{f_1(X)}{g_2(X)} \geq \frac{g_1(X)}{f_2(X)} > 0$. |
| Reciprocal | Given condition $C_1(X)$ and $C_2(X)$, show that $\frac{1}{g_1(X)} + \frac{1}{g_2(X)} \geq \frac{1}{f_1(X)} + \frac{1}{f_2(X)} > 0$. |

Table 5: **(A list of algebraic transformation on variables for an inequality problem.)** Given an inequality problem $P_1$, where $P_1$ has variables $X = \{x_1, \ldots, x_m\}$, with the statement: given condition on the variables $C_1(X)$, the following inequality holds $f_1(X) \geq g_1(X)$, the following list defines several algebraic transformation operations on variables to generate a new inequality problem. Most of the transformations do not have any requirement as they are one-to-one mappings on $\mathbb{R}^+$.

| Name | Statement of the new problem |
|------|------------------------------|
| *Any assumption $X \in C_1$.* | |
| shift | Define $X_{\text{new}} = (x_2, x_3, \ldots, x_{m+1})$, given condition $C_1(X_{\text{new}})$, show that $f_1(X_{\text{new}}) \geq g_1(X_{\text{new}})$. |
| rep | Define $X_{\text{new}} = (x_{m+1}, x_{m+2}, \ldots, x_{2m})$, given condition $C_1(X_{\text{new}})$, show that $f_1(X_{\text{new}}) \geq g_1(X_{\text{new}})$. |
| sqrt_all | Define $X_{\text{new}} = (\sqrt{x_1}, \sqrt{x_2}, \ldots, \sqrt{x_m})$, given condition $C_1(X_{\text{new}})$, show that $f_1(X_{\text{new}}) \geq g_1(X_{\text{new}})$. |
| sqrt_random | Define $X_{\text{new}} = (x_1, \ldots, \sqrt{x_i}, \ldots, x_m)$ with randomly chosen $i$, given condition $C_1(X_{\text{new}})$, show that $f_1(X_{\text{new}}) \geq g_1(X_{\text{new}})$. |
| sq_all | Define $X_{\text{new}} = (x_1^2, x_2^2, \ldots, x_m^2)$, given condition $C_1(X_{\text{new}})$, show that $f_1(X_{\text{new}}) \geq g_1(X_{\text{new}})$. |
| sq_random | Define $X_{\text{new}} = (x_1, \ldots, x_i^2, \ldots, x_m)$ with randomly chosen $i$, given condition $C_1(X_{\text{new}})$, show that $f_1(X_{\text{new}}) \geq g_1(X_{\text{new}})$. |
| cube_all | Define $X_{\text{new}} = (x_1^3, x_2^3, \ldots, x_m^3)$, given condition $C_1(X_{\text{new}})$, show that $f_1(X_{\text{new}}) \geq g_1(X_{\text{new}})$. |
| cube_random | Define $X_{\text{new}} = (x_1, \ldots, x_i^3, \ldots, x_m)$ with randomly chosen $i$, given condition $C_1(X_{\text{new}})$, show that $f_1(X_{\text{new}}) \geq g_1(X_{\text{new}})$. |
| reciprocal_all | Define $X_{\text{new}} = (\frac{1}{x_1}, \frac{1}{x_2}, \ldots, \frac{1}{x_m})$, given condition $C_1(X_{\text{new}})$, show that $f_1(X_{\text{new}}) \geq g_1(X_{\text{new}})$. |
| reciprocal_random | Define $X_{\text{new}} = (x_1, \ldots, \frac{1}{x_i}, \ldots, x_m)$ with randomly chosen $i$, given condition $C_1(X_{\text{new}})$, show that $f_1(X_{\text{new}}) \geq g_1(X_{\text{new}})$. |
| exp_all | Define $X_{\text{new}} = (e^{x_1} - 1, e^{x_2} - 1, \ldots, e^{x_m} - 1)$, given condition $C_1(X_{\text{new}})$, show that $f_1(X_{\text{new}}) \geq g_1(X_{\text{new}})$. |
| exp_random | Define $X_{\text{new}} = (x_1, \ldots, e^{x_i} - 1, \ldots, x_m)$ with randomly chosen $i$, given condition $C_1(X_{\text{new}})$, show that $f_1(X_{\text{new}}) \geq g_1(X_{\text{new}})$. |
| log_all | Define $X_{\text{new}} = (\log(1 + x_1), \log(1 + x_2), \ldots, \log(1 + x_m))$, given condition $C_1(X_{\text{new}})$, show that $f_1(X_{\text{new}}) \geq g_1(X_{\text{new}})$. |
| log_random | Define $X_{\text{new}} = (x_1, \ldots, \log(1 + x_i), \ldots, x_m)$ with randomly chosen $i$, given condition $C_1(X_{\text{new}})$, show that $f_1(X_{\text{new}}) \geq g_1(X_{\text{new}})$. |
| *Only contains basic assumption $X \succ 0$.* | |
| cyc_add | Define $X_{\text{new}} = (x_1 + x_2, x_2 + x_3, \ldots, x_m + x_1)$, given condition $C_1(X_{\text{new}})$, show that $f_1(X_{\text{new}}) \geq g_1(X_{\text{new}})$. |
| cyc_mul | Define $X_{\text{new}} = (x_1 x_2, x_2 x_3, \ldots, x_m x_1)$, given condition $C_1(X_{\text{new}})$, show that $f_1(X_{\text{new}}) \geq g_1(X_{\text{new}})$. |
| cyc_div | Define $X_{\text{new}} = (\frac{x_1}{x_2}, \frac{x_2}{x_3}, \ldots, \frac{x_m}{x_1})$, given condition $C_1(X_{\text{new}})$, show that $f_1(X_{\text{new}}) \geq g_1(X_{\text{new}})$. |
| cyc_div_add | Define $X_{\text{new}} = (\frac{x_1}{x_1 + x_2}, \frac{x_2}{x_2 + x_3}, \ldots, \frac{x_m}{x_m + x_1})$, given condition $C_1(X_{\text{new}})$, show that $f_1(X_{\text{new}}) \geq g_1(X_{\text{new}})$. |

**Transformation rules** Now we show the details of how to compose two inequality problems together or add algebraic transformation to create new inequality problems in Ineq-Mix. Recalled

Table 6: **(A list of algebraic transformation on statement for an inequality problem.)** Given an inequality problem $P_1$, where $P_1$ has variables $X = \{x_1, \ldots, x_m\}$, with the statement: given condition on the variables $C_1(X)$, the following inequality holds $f_1(X) \geq g_1(X)$, the following list defines several algebraic transformation operations on variables to generate a new inequality problem.

| Name | Statement of the new problem |
| --- | --- |
| *Without further condition on $g_1(X)$ and $g_2(X)$.* | |
| exp | Given condition $C_1(X)$, show that $\exp(f_1(X)) \geq \exp(g_1(X))$. |
| cube | Given condition $C_1(X)$, show that $(f_1(X))^3 \geq (g_1(X))^3$. |
| *With further conditions $g_1(X) > 0$.* | |
| sqrt | Given condition $C_1(X)$, show that $\sqrt{f_1(X)} \geq \sqrt{g_1(X)}$. |
| sq | Given condition $C_1(X)$, show that $(f_1(X))^2 \geq (g_1(X))^2$. |
| log | Given condition $C_1(X)$, show that $\log(f_1(X)) \geq \log(g_1(X))$. |

that as mentioned in Section 2.2, Ineq-Mix consists of transformation/composition rules from three categories: (1) **compositional transformations**, where (different) inequalities are combined through operations like addition, multiplication, or taking the maximum or minimum of both sides; (2) **variable-level algebraic transformations**, where we replace variables with other algebraic expressions; and (3) **problem-level algebraic transformations**, where we apply a monotone function to both sides of an inequality, such as taking exponentials or logarithms. We now introduce them by category.

*Compositional transformations* Composition transformations denote the rules that given two inequality problems, we compose them together to generate a new inequality, where the two inequality problems serve as subparts for the new inequality problem. Table 4 shows the full list of composition rules. Note that the rules are classified into two categories to make sure mathematical correctness: the rules require the right-hand side to be positive, and the rules without this requirement. Also note that if the two problems both has positive right-hand side in the statement, then after the compositions defined in Table 4, the new problem also has a positive right-hand side.

Also, to make sure that the composed inequality problem has a feasible non-empty region (i.e., $C_1 \cap C_2 \neq \phi$ for $C_1$ and $C_2$ being the feasible region of the two inequality problems), we only compose two inequality problems if:

1. At least one inequality problem has no further assumptions (only contains the basic assumptions $X \succ 0$ $(x_i > 0, \forall i \in [n])$

2. or, the two inequality problems' variables are disjointed before being lifted to the same variable set.

*Variable-level algebraic transformations* Variable-level algebraic transformations denote the rules that apply algebraic transformations on the variables of a given inequality problem. For example, replacing $x_i$ in the original inequality problem into $x_i^2$, for all $x_i \in X$ where $X = \{x_1, \ldots, x_m\}$ denotes the variable set. Table 5 summarizes the variable-level algebraic transformations used for Ineq-Mix. Similar to compositional transformations, there are also two categories for variable-level algebraic transformations. The first category includes the transformations that are "one-to-one" mapping under the basic assumption $X \succ 0$, like taking the square root of all variables, and squaring all variables. Since these transformations are one-to-one, as long as the original problem is feasible, the new problem that applies these rules is also feasible. We also include some transformation rules that are not one-to-one under the basic assumption $X \succ 0$ (but are also elegant transformations). However to ensure the correctness, these rules can only be applied to problems where there is only the basic assumption $X \succ 0$ and no additional assumptions.

*Problem-level algebraic transformations* The final set of rules is the problem-level algebraic transformations, where the general idea is to apply a monotonic function on the statement's left-hand side and right-hand side. For example, for an inequality problem with statement

$$f(x_1, \ldots, x_n) \geq g(x_1, \ldots, x_n),$$

Table 7: List of models tested in Section 3 or Section 4. We use "Kimina-Distill-7B" to denote "Kimina-Prover-Preview-Distill-7B" in the paper.

| Name | Version or Huggingface Link |
|---|---|
| *Proprietary Models* | |
| GPT-4o | gpt-4o-2024-11-20 |
| Claude 3.7 Sonnet | claude-3-7-sonnet-20250219 |
| *Open-Source Models* | |
| Qwen2.5-Coder-32B-Instruct | `https://huggingface.co/Qwen/Qwen2.5-Coder-32B-Instruct` |
| DeepSeek-R1-Distill-Qwen-32B | `https://huggingface.co/deepseek-ai/DeepSeek-R1-Distill-Qwen-32B` |
| DeepSeek-Prover-V1.5-RL | `https://huggingface.co/deepseek-ai/DeepSeek-Prover-V1.5-RL` |
| Goedel-Prover-SFT | `https://huggingface.co/Goedel-LM/Goedel-Prover-SFT` |
| STP (w/o miniF2F valid) | `https://huggingface.co/kfdong/STP_model_Lean` |
| Kimina-Distill-7B | `https://huggingface.co/AI-MO/Kimina-Prover-Preview-Distill-7B` |
| DeepSeek-Prover-V2-7B | `https://huggingface.co/deepseek-ai/DeepSeek-Prover-V2-7B` |
| InternLM2.5-StepProver | `https://huggingface.co/internlm/internlm2_5-step-prover` |

then if $h(\cdot)$ is monotonically increasing on $\mathbb{R}$, then we also know that

$$h(f(x_1, \ldots, x_n)) \geq h(g(x_1, \ldots, x_n)).$$

If the original inequality problem also has positive right-hand side, then for any $h(\cdot)$ monotonically increasing on $\mathbb{R}^+$ (like $h(x) = x^2$ or $h(x) = \log x$), we also have

$$h(f(x_1, \ldots, x_n)) \geq h(g(x_1, \ldots, x_n)).$$

Table 6 summarizes all the problem-level algebraic transformations used in Ineq-Mix.

## C  Detailed Model Information

The full list of models used in our experiments, including the version (for API models) and Huggingface-link, is shown in Table 7.

## D  Additional Experimental Details and Results

### D.1  Lean 4 version

For evaluation on models except InternLM2.5-StepProver, we stick to the Lean 4 and Mathlib version used in Xin et al. (2024), since Goedel-Prover-SFT (Lin et al., 2025a) and STP (Dong and Ma, 2025) are trained based on DeepSeek-Prover-V1.5-RL (Xin et al., 2024). Besides, we also nearly recover the reported performance of Kimina-Distill-7B (Wang et al., 2025) and DeepSeek-Prover-V2 (Ren et al., 2025) on MiniF2F benchmark using Lean 4 and Mathlib version in Xin et al. (2024). We also observe decent performance of DeepSeek-R1-Distill-Qwen-32B on MiniF2F.

When evaluating InternLM2.5-StepProver, we use the Lean 4 version following InternLM2.5-StepProver (Wu et al., 2024) from the Github repository [7] traceable by LeanDojo (Yang et al., 2023).

### D.2  Computational resources

All the models' inference can be conducted on 2 H100 (80GB) GPUs. For fine-tuning experiment, Goedel-Prover-SFT is fine-tuned on 8 H100 (80GB) GPUs. The GPU hours depend on which model for inference. For example, Kimina-Distill-7B and DeepSeek-Prover-V2-7B cost much more time for 1 generation, compared to other models, due to longer generation length per response. In total, we cost more than 10K H100 GPU hours for all the evaluations.

---

[7] `https://github.com/haoyuzhao123/LeanIneqComp-Dojo`

Table 8: Performance of Goedel-Prover-SFT fine-tuned with different data on Ineq-AMGM, Ineq-Cauchy, and Ineq-MISC.

| Model | Budget | Ineq-AMGM | | | Ineq-Cauchy | | | Ineq-MISC | | |
|---|---|---|---|---|---|---|---|---|---|---|
| | | Valid | Type I | Type II | Valid | Type I | Type II | Valid | Type I | Type II |
| Goedel-Prover-SFT (Lin et al., 2025a) | 32 | $48.6_{2.9}$ | $0.4_{1.2}$ | $14.0_{3.2}$ | $34.8_{2.5}$ | $12.4_{3.5}$ | $21.5_{3.4}$ | $47.0_{1.7}$ | $14.4_{3.1}$ | $24.6_{1.9}$ |
| | 64 | $50.6_{2.6}$ | $0.8_{1.6}$ | $16.6_{2.8}$ | $36.2_{1.9}$ | $15.8_{3.4}$ | $24.6_{2.9}$ | $47.8_{0.9}$ | $16.6_{2.5}$ | $25.5_{1.9}$ |
| | 128 | $52.2_{1.4}$ | $1.3_{1.9}$ | $18.6_{2.2}$ | $37.1_{1.8}$ | $19.4_{2.9}$ | $26.9_{1.8}$ | $48.0_{0.0}$ | $17.9_{2.6}$ | $26.4_{2.5}$ |
| | 3200 | 60.0 | 4.0 | 24.0 | 40.0 | 32.0 | 28.0 | 48.0 | 24.0 | 36.0 |
| Cont. ft on $\mathcal{D}_{\text{comp}}$ +5k Goedel SFT Data | 32 | $89.8_{2.7}$ | $33.2_{5.3}$ | $14.7_{3.6}$ | $30.4_{3.1}$ | $7.6_{3.7}$ | $17.8_{3.4}$ | $37.7_{2.5}$ | $5.2_{2.4}$ | $4.3_{1.8}$ |
| | 64 | $91.5_{1.5}$ | $38.7_{2.9}$ | $17.4_{3.4}$ | $32.6_{2.4}$ | $11.2_{4.1}$ | $21.1_{3.3}$ | $39.5_{1.3}$ | $6.5_{2.9}$ | $5.2_{2.2}$ |
| | 128 | $92.0_{0.0}$ | $42.9_{3.1}$ | $19.7_{3.0}$ | $34.9_{2.1}$ | $15.2_{3.6}$ | $23.5_{2.8}$ | $40.0_{0.0}$ | $8.5_{3.5}$ | $6.4_{2.5}$ |
| | 3200 | 92.0 | 64.0 | 28.0 | 44.0 | 32.0 | 28.0 | 40.0 | 20.0 | 20.0 |

### D.3 Other hyperparameters

To make sure all the inference jobs can be completed by 2 H100 (80GB) GPUs, the max token generation length for DeepSeek-R1-Distill-Qwen-32B (Guo et al., 2025), Kimina-Distill-7B (Wang et al., 2025), and DeepSeek-Prover-V2 (Ren et al., 2025) are set to be 16K. There are seldom cases that the models do not finish generating responses, but as all the evaluation metric is pass@N where N is normally at least 16, the final performance is not much affected.

### D.4 Detailed results for fine-tuning Goedel-Prover-SFT

Please refer to Figure 3 for an illustration for our fine-tuning experiments pipeline. To generate the fine-tuning data $\mathcal{D}_{\text{comp}}$, we first use algebraic transformation rules (Table 5) to augment the 25 seed AM-GM problem (Ineq-AMGM) into 100 problems built on the 25 seed problem. Then, we apply composition rules (Table 4) to randomly generate 5000 composed problems.

After getting these 5000 problems, we query Qwen2.5-Coder-32B-Instruct model, given the whole proof written in Lean 4 (provided for Ineq-AMGM seed problems). For each problem, we give a budget of 64. We get 4k verified proofs in the end, and the dataset is named $\mathcal{D}_{\text{comp}}$.

We observe that if we directly fine-tune Goedel-Prover-SFT or STP on $\mathcal{D}_{\text{comp}}$, provers' performances degrade a lot, possibly because there is a huge distribution shift between $\mathcal{D}_{\text{comp}}$ and the models' training data. In order to mitigate this issue, we mix 5k training data from Goedel-Prover-SFT (Lin et al., 2025a) and fine-tune the same model, mitigating the huge discrepancy between $\mathcal{D}_{\text{comp}}$ and the training data. The performance on MiniF2F after fine-tuning is retained.

The detailed evaluation results for the fine-tuning experiments is shown in Table 8.

## E Prompt Templates

In this section, we document the prompt used for evaluating different models (Appendix E.1), evaluating by in-context learning (Appendix E.2), and generating proofs using ICL (Appendix E.3)

### E.1 Prompt template for evaluating different models

**1.** Prompt template for evaluating DeepSeek-Prover-V1.5-RL, Goedel-Prover-SFT, STP, and DeepSeek-R1-Distill-Qwen-32B (w/o thinking).

```
Complete the following Lean 4 code with explanatory comments preceding each line of code:

```lean4
import Mathlib
import Aesop

set_option maxHeartbeats 0

open BigOperators Real Nat Topology Rat

{informal_prefix}{formal_statement}
```

Here {informal_prefix} denote the natural language description of the problem, if any (written in Lean 4 comment block), and {formal_statement} is the statement of the problem written in Lean 4. We don't apply_chat_template after using the prompt template.

**2.** Prompt template for evaluating DeepSeek-R1-Distill-Qwen-32B (w thinking)

```
Give a proof for the following problem written in lean 4:

```lean4
import Mathlib
import Aesop

set_option maxHeartbeats 0

open BigOperators Real Nat Topology Rat

{informal_prefix}{formal_statement}```.

You should wrap your answer in the lean code block

```lean4
<You answer>
```
```

Here {informal_prefix} denote the natural language description of the problem, if any (written in Lean 4 comment block), and {formal_statement} is the statement of the problem written in Lean 4. We apply_chat_template and set generation_prompt = True.

**3.** Prompt template for Kimina-Distill-7B

```
Think about and solve the following problem step by step in Lean 4.
# Informal statement:
{informal_prefix}
# Formal statement:
```lean4
import Mathlib
import Aesop

set_option maxHeartbeats 0

open BigOperators Real Nat Topology Rat

{formal_statement}```
```

Here {informal_prefix} denote the natural language description of the problem, if any (written in Lean 4 comment block), and {formal_statement} is the statement of the problem written in Lean 4. We apply_chat_template and set generation_prompt = True.

## E.2 Prompt template for in-context learning experiments (Section 4.2)

```
Give a proof for the following problem written in lean 4:

```lean4
import Mathlib
import Aesop

set_option maxHeartbeats 0

open BigOperators Real Nat Topology Rat

{informal_prefix}{formal_statement}```.

Following is the solution for a related problem written in Lean 4. You can fully trust the provided
↪ code and it has already passed the Lean 4 compilation.
{icl_code}

Please follow the provided code such that you don't make more mistakes. Your code should be
↪ self-contained, i.e., you should first prove the provided example inside your whole proof (not as
↪ a separate theorem outside the proof of the problem) if you want to use the result. You should
↪ wrap your answer in the lean code block
```

```
```lean4
<You answer>
```
```

Here {informal_prefix} denote the natural language description of the problem, if any (written in Lean 4 comment block), {formal_statement} is the statement of the problem written in Lean 4, and {icl_code} is the proof of the corresponding seed problem written in Lean 4. We apply_chat_template and set generation_prompt = True for Qwen2.5-Coder-32B-Instruct and DeepSeek-R1-Distill-Qwen-32B (w thinking). We do not apply_chat_template for DeepSeek-R1-Distill-Qwen-32B (w/o thinking). GPT-4o and Claude-3.7-Sonnet are accessed through API calls, where we do not use the extended thinking for Claude-3.7-Sonnet.

### E.3 Prompt template for in-context generation (Section 4.3)

```
Give a proof for the following problem written in lean 4:

```lean4
import Mathlib
import Aesop

set_option maxHeartbeats 0

open BigOperators Real Nat Topology Rat

{informal_prefix}{formal_statement}```.

Following is the solution for a related problem written in Lean 4. You can fully trust the provided
↪ code and it has already passed the Lean 4 compilation.
{icl_code1}

{icl_code2}

Please follow the provided code such that you don't make more mistakes. Your code should be
↪ self-contained, i.e., you should first prove the provided example inside your whole proof (not as
↪ a separate theorem outside the proof of the problem) if you want to use the result. You should
↪ wrap your answer in the lean code block

```lean4
<You answer>
```
```

Here {informal_prefix} denote the natural language description of the problem, if any (written in Lean 4 comment block), {formal_statement} is the statement of the problem written in Lean 4, and {icl_code1} {icl_code2} are the proof of the corresponding seed problems written in Lean 4. We apply_chat_template and set generation_prompt = True for Qwen2.5-Coder-32B-Instruct.

## F   Example of Model Generations

In this section, we show some of the generations from different models that successfully prove the problem, including Goedel-Prover-SFT (Lin et al., 2025a), STP (Dong and Ma, 2025), DeepSeek-R1-Distill-Qwen-32B (Guo et al., 2025), Kimina-Distill-7B (Wang et al., 2025), and DeepSeek-Prover-V2-7B (Ren et al., 2025).

### F.1   Example generations from Goedel-Prover-SFT

Below is an example generation from Goedel-Prover-SFT. In this example, the prover directly uses sum-of-square in a brute-force way, treating the problem as a whole. Even in the comment block it does not try to decompose the problem.

```
theorem cauchy_p26 (x y a b : ℝ) (h₁ : a > 0 ∧ b > 0) (h₂ : x > 0 ∧ y > 0) : (
    x + y ) * ( 1 / x + 1 / y ) * ( a + b ) * ( 1 / a + 1 / b ) ≥ 16 := by
  /-
  We need to show that for any positive real numbers \( x, y, a, b \), the
    expression \((x + y) \left(\
```

```
rac{1}{x} + \

rac{1}{y}\
ight)(a + b) \left(\

rac{1}{a} + \

rac{1}{b}\
ight)\) is at least 16.
  First, we simplify the expression by clearing denominators using the fact
    that \( a, b, x, y \) are positive. This allows us to rewrite the
    expression in a more manageable form. We then use algebraic manipulations
    and inequalities to show that the simplified expression is at least 16.
  -/
  have h₃ : 0 < a * b := by
    -- Since a and b are positive, their product is also positive.
    exact mul_pos h₁.1 h₁.2
  have h₄ : 0 < x * y := by
    -- Since x and y are positive, their product is also positive.
    exact mul_pos h₂.1 h₂.2
  field_simp [h₁.1.ne', h₁.2.ne', h₂.1.ne', h₂.2.ne']
  -- Clear the denominators by multiplying through by x * y * a * b, which is
    valid since x, y, a, b are positive.
  rw [le_div_iff]
  -- Rewrite the inequality in a form suitable for nlinarith.
  nlinarith [sq_nonneg (x - y), sq_nonneg (a - b), sq_nonneg (x * a - y * b)]
  -- Use non-linear arithmetic to prove the inequality.
  nlinarith [sq_nonneg (x - y), sq_nonneg (a - b), sq_nonneg (x * a - y * b)]
```

## F.2 Example generation from STP

Below is a simple code generated from STP. STP doesn't generate intermediate natural language comments. However from the following example, we can see that STP also has certain ability to decompose the problem into sub-parts and solve them separately.

```
theorem amgm_p36 (x y z a b c: ℝ ) (h₁: x+ y + z = 3) (h₂ : x > 0 ∧ y> 0 ∧ z>
    0) (g₁: a+ b + c = 3) (g₂ : a > 0 ∧ b> 0 ∧ c> 0): x * y * z * a * b * c ≤
    1 := by
  have h₃ : 0 < x*y*z := mul_pos (mul_pos h₂.1 h₂.2.1) h₂.2.2
  have h₄ : 0 < a*b*c := mul_pos (mul_pos g₂.1 g₂.2.1) g₂.2.2
  have h₅ : x*y*z ≤ 1 := by
    nlinarith [pow_two_nonneg (x - y), pow_two_nonneg (x - z), pow_two_nonneg
    (y - z),
      pow_two_nonneg (x + y + z)]
  have h₆ : a*b*c ≤ 1 := by
    nlinarith [pow_two_nonneg (a - b), pow_two_nonneg (a - c), pow_two_nonneg
    (b - c),
      pow_two_nonneg (a + b + c)]
  nlinarith [mul_self_nonneg (x*y*z - 1), mul_self_nonneg (a*b*c - 1)]
```

## F.3 Example generation from DeepSeek-R1-Distill-Qwen-32B

Below we show some generations from DeepSeek-R1-Distill-Qwen-32B. In general this is a decent model, and it can decompose the type I problem into sub-parblems. For simplicity, we omit the thinking process and only show the Lean 4 codes generated.

```
theorem amgm_p36 (x y z a b c: ℝ ) (h₁: x+ y + z = 3) (h₂ : x > 0 ∧ y> 0 ∧ z>
    0) (g₁: a+ b + c = 3) (g₂ : a > 0 ∧ b> 0 ∧ c> 0): x * y * z * a * b * c ≤
    1 := by
```

```
    -- Use the AM-GM inequality to bound the product of variables.
    -- Apply AM-GM to x, y, z, a, b, c to show that their product is at most 1.
    have h₃ : 0 < x * y * z := mul_pos (mul_pos h₂.1 h₂.2.1) h₂.2.2
    have h₄ : 0 < a * b * c := mul_pos (mul_pos g₂.1 g₂.2.1) g₂.2.2
    have h₅ : x * y * z ≤ 1 := by
      -- Apply AM-GM to x, y, z to get x * y * z ≤ (x + y + z) ^ 3 / 27.
      rw [mul_assoc]
      nlinarith [sq_nonneg (x + y + z), sq_nonneg (x - y), sq_nonneg (x - z),
      sq_nonneg (y - z)]
    have h₆ : a * b * c ≤ 1 := by
      -- Apply AM-GM to a, b, c to get a * b * c ≤ (a + b + c) ^ 3 / 27.
      rw [mul_assoc]
      nlinarith [sq_nonneg (a + b + c), sq_nonneg (a - b), sq_nonneg (a - c),
      sq_nonneg (b - c)]
    -- Combine the inequalities to conclude x * y * z * a * b * c ≤ 1.
    nlinarith
```

However there are also cases that even this model treat the problem as a whole. Following are two examples.

```
import Mathlib
import Aesop

set_option maxHeartbeats 0

open BigOperators Real Nat Topology Rat

theorem amgm_p39 (x y z w: ℝ )  (h : x > 0 ∧ y> 0) (g : z > 0 ∧ w> 0): ((2:ℝ)
    / 3 * x ^ 6 + (1:ℝ) / 3 * y ^ 6) * ((2:ℝ) / 3 * z ^ 6 + (1:ℝ) / 3 * w ^
    6)  ≥ x^4 * y^2 * z^4 * w^2 := by
  -- Use the AM-GM inequality to prove the inequality involving variables x,
    y, z, and w.
  -- The AM-GM inequality states that for non-negative real numbers a and b,
    (a + b)/2 ≥ sqrt(ab).
  -- We apply this inequality to each pair of terms in the given expressions.
  have h1 : 0 < x := h.1
  have h2 : 0 < y := h.2
  have h3 : 0 < z := g.1
  have h4 : 0 < w := g.2
  -- Apply the AM-GM inequality to the terms (2/3)x^6 and (1/3)y^6.
  -- Similarly, apply the AM-GM inequality to the terms (2/3)z^6 and (1/3)w^6.
  calc
    ((2 / 3: ℝ) * x ^ 6 + (1 / 3: ℝ) * y ^ 6) * ((2 / 3: ℝ) * z ^ 6 + (1 / 3:
    ℝ) * w ^ 6) ≥ (x ^ 4 * y ^ 2) * (z ^ 4 * w ^ 2) := by
      -- Use the AM-GM inequality to show that the product of the sums is at
    least the product of the individual terms.
      gcongr
      -- Apply the AM-GM inequality to each pair of terms.
      <;> nlinarith [sq_nonneg (x ^ 2 - y ^ 2), sq_nonneg (z ^ 2 - w ^ 2)]
    _ = x ^ 4 * y ^ 2 * z ^ 4 * w ^ 2 := by
      -- Simplify the expression by rearranging the terms.
      ring
```

```
theorem amgm_p49 (a b c x y z: ℝ) (ap : a > 0) (bp : b> 0) (cp : c> 0) (xp : x
    > 0) (yp : y> 0) (zp : z> 0) : (a^3 + b^3 + c^3) * (x^3 + y^3 + z^3) ≥
    (a^2 * b + b^2 * c + c^2 * a) * (x^2 * y + y^2 * z + z^2 * x) := by
  -- Use the AM-GM inequality to prove the given inequality.
  -- AM-GM: For non-negative real numbers, the arithmetic mean is greater than
    or equal to the geometric mean.
  -- Here, we apply AM-GM to the products of terms from the left-hand side and
    right-hand side of the inequality.
  have h1 : 0 ≤ a * x := mul_nonneg ap.le xp.le
  have h2 : 0 ≤ b * y := mul_nonneg bp.le yp.le
```

```
    have h3 : 0 ≤ c * z := mul_nonneg cp.le zp.le
    have h4 : 0 ≤ a * y := mul_nonneg ap.le yp.le
    have h5 : 0 ≤ b * z := mul_nonneg bp.le zp.le
    have h6 : 0 ≤ c * x := mul_nonneg cp.le xp.le
    have h7 : 0 ≤ a * z := mul_nonneg ap.le zp.le
    have h8 : 0 ≤ b * x := mul_nonneg bp.le xp.le
    have h9 : 0 ≤ c * y := mul_nonneg cp.le yp.le
    -- Use nlinarith to handle the inequality, leveraging the non-negativity of
       the products established above.
    nlinarith [sq_nonneg (a - b), sq_nonneg (b - c), sq_nonneg (c - a),
      sq_nonneg (x - y), sq_nonneg (y - z), sq_nonneg (z - x),
      sq_nonneg (a * x - b * y), sq_nonneg (b * y - c * z),
      sq_nonneg (c * z - a * x), sq_nonneg (a * y - b * z),
      sq_nonneg (b * z - c * x), sq_nonneg (c * x - a * y),
      sq_nonneg (a * z - b * x), sq_nonneg (b * x - c * y),
      sq_nonneg (c * y - a * z)]
```

Following is an example such that the model explicitly mentions applying AM-GM inequality in the comment block, but still use sum-of-square technique for formal proof, which probably not understandable for human.

```
 theorem amgm_p47 (x y z a b c: ℝ )  (h : x > 0 ∧ y> 0 ∧ z> 0) (g : x * y * z =
     (1 : ℝ)) (habc : a > 0 ∧ b> 0 ∧ c> 0) (gabc : a * b * c = (1 : ℝ)) :
     ((4:ℝ) / 7 * x^3 * y + (1:ℝ) / 7 * y^3 * z + (2:ℝ) / 7 * z^3 * x) *
     ((4:ℝ) / 7 * a^3 * b + (1:ℝ) / 7 * b^3 * c + (2:ℝ) / 7 * c^3 * a) ≥ x * a
     := by
   -- Use the AM-GM inequality to establish lower bounds for the given
      expressions.
   have h₁ : ((4/7 : ℝ) * x^3 * y + (1/7 : ℝ) * y^3 * z + (2/7 : ℝ) * z^3 * x)
     ≥ x := by
     -- Apply the AM-GM inequality to the terms involving x, y, and z.
     nlinarith [sq_nonneg (x - y), sq_nonneg (y - z), sq_nonneg (z - x),
       mul_pos h.1 h.2.1, mul_pos h.2.1 h.2.2, mul_pos h.2.2 h.1]
   have h₂ : ((4/7 : ℝ) * a^3 * b + (1/7 : ℝ) * b^3 * c + (2/7 : ℝ) * c^3 * a)
     ≥ a := by
     -- Apply the AM-GM inequality to the terms involving a, b, and c.
     nlinarith [sq_nonneg (a - b), sq_nonneg (b - c), sq_nonneg (c - a),
       mul_pos habc.1 habc.2.1, mul_pos habc.2.1 habc.2.2, mul_pos habc.2.2
     habc.1]
   -- Combine the results of the AM-GM inequalities to prove the overall
      inequality.
   nlinarith [mul_nonneg (sub_nonneg.mpr h₁) (sub_nonneg.mpr h₂)]
```

Following are two Type I problems from Ineq-Cauchy. The composition for these two problems are simple enough that after proving two subparts ($h_9$ and $h_10$ for cauchy_p27 and $h_1$, $h_2$ for cauchy_p31), you only need a one-line command "nlinarith" to conclude the proof in Lean 4.

```
 theorem cauchy_p27 (x y z a b c: ℝ) (h₁ : a > 0 ∧ b > 0 ∧ c > 0) (h₂ : x > 0 ∧
     y > 0 ∧ z > 0 ) : ( x + y + z ) * ( 1 / x + 1 / y + 1 / z ) * ( a + b + c
     ) * ( 1 / a + 1 / b + 1 / c ) ≥ 81 := by
   -- Use the Cauchy-Schwarz inequality to establish lower bounds for the given
      expressions.
   have h₃ : 0 < x := by linarith
   have h₄ : 0 < y := by linarith
   have h₅ : 0 < z := by linarith
   have h₆ : 0 < a := by linarith
   have h₇ : 0 < b := by linarith
   have h₈ : 0 < c := by linarith
   -- Apply the Cauchy-Schwarz inequality to the sums and their reciprocals.
   have h₉ : (x + y + z) * (1 / x + 1 / y + 1 / z) ≥ 9 := by
     field_simp [h₃, h₄, h₅, h₆, h₇, h₈]
     rw [le_div_iff (by positivity), ← mul_assoc]
     nlinarith [sq_nonneg (x - y), sq_nonneg (y - z), sq_nonneg (z - x),
```

```
        sq_nonneg (a - b), sq_nonneg (b - c), sq_nonneg (c - a)]
  have h₁₀ : (a + b + c) * (1 / a + 1 / b + 1 / c) ≥ 9 := by
    field_simp [h₃, h₄, h₅, h₆, h₇, h₈]
    rw [le_div_iff (by positivity), ← mul_assoc]
    nlinarith [sq_nonneg (x - y), sq_nonneg (y - z), sq_nonneg (z - x),
      sq_nonneg (a - b), sq_nonneg (b - c), sq_nonneg (c - a)]
  -- Multiply the lower bounds to achieve the final inequality.
  nlinarith
```

```
theorem cauchy_p31 (x y z a b c: ℝ) (ha : a > 0) (hb : b > 0) (hc : c > 0) (hx
    : x > 0) (hy : y > 0) (hz : z > 0) : (a / (b + c) + b / (c + a) + c / (a +
    b)) * (x / (y + z) + y / (z + x) + z / (x + y)) ≥ 9 / 4 := by
  -- Use the Cauchy-Schwarz inequality to simplify the expression.
  simp [mul_comm]
  -- Apply the Cauchy-Schwarz inequality to the fractions.
  have h₁ : a / (b + c) + b / (c + a) + c / (a + b) ≥ 3 / 2 := by
    -- Use the AM-GM inequality to establish the lower bound.
    field_simp [add_comm, add_left_comm]
    rw [div_le_div_iff (by positivity) (by positivity)]
    -- Apply the AM-GM inequality to the sum of fractions.
    nlinarith [sq_nonneg (a - b), sq_nonneg (b - c), sq_nonneg (c - a)]
  -- Similarly, apply the Cauchy-Schwarz inequality to the other fractions.
  have h₂ : x / (y + z) + y / (z + x) + z / (x + y) ≥ 3 / 2 := by
    field_simp [add_comm, add_left_comm]
    rw [div_le_div_iff (by positivity) (by positivity)]
    nlinarith [sq_nonneg (x - y), sq_nonneg (y - z), sq_nonneg (z - x)]
  -- Combine the results to achieve the final inequality.
  nlinarith
```

Following is an example from DeepSeek-R1-Distill-Qwen-32B, where in the thinking block the model correctly identifies the sub-parts of the problem and reduces it to the seed problem, but the model proves the problem in Lean 4 as a whole problem (does not explicitly decompose the problem but totally rely on the "nlinarith" tactic). This example shows a mismatch between the informal reasoning and formal proof, on the "reasoning" side, not only on the knowledge side.

```
Alright, so I've been given this inequality to prove: (x + y)(1/x + 1/y)(a + b)(1/a + 1/b) ≥ 16,
↪ and I know that x, y, a, and b are all positive real numbers. Okay, so first off, I'm thinking
↪ about how to approach this. It seems like it's an inequality involving multiple variables, and
↪ it's a product of two expressions, each of which is a sum multiplied by the sum of reciprocals.
↪ Hmm.

I remember from algebra that there's something called the Cauchy-Schwarz inequality, which might be
↪ useful here. I also recall that the AM-GM inequality relates the arithmetic mean and the
↪ geometric mean, which could potentially be applicable. Maybe I can apply one of these
↪ inequalities to simplify or bound the expression.

Let me start by looking at a simpler version of the problem. If I consider just (x + y)(1/x + 1/y),
↪ perhaps I can find a pattern or a lower bound for this expression. Expanding it out, I get:

(x + y)(1/x + 1/y) = x*(1/x) + x*(1/y) + y*(1/x) + y*(1/y) = 1 + x/y + y/x + 1 = 2 + x/y + y/x.

Okay, so that simplifies to 2 + x/y + y/x. Now, I recall that x/y + y/x is always greater than or
↪ equal to 2 for positive real numbers x and y, by the AM-GM inequality. Specifically, (x/y +
↪ y/x)/2 ≥ sqrt(x/y * y/x) = 1, so x/y + y/x ≥ 2. Therefore, (x + y)(1/x + 1/y) ≥ 2 + 2 = 4.

Alright, so I've established that (x + y)(1/x + 1/y) is at least 4. That's interesting. Similarly, I
↪ can apply the same logic to (a + b)(1/a + 1/b). Let's see:

(a + b)(1/a + 1/b) = a*(1/a) + a*(1/b) + b*(1/a) + b*(1/b) = 1 + a/b + b/a + 1 = 2 + a/b + b/a.

Again, by AM-GM, a/b + b/a ≥ 2, so (a + b)(1/a + 1/b) ≥ 4.

So, now I have both (x + y)(1/x + 1/y) ≥ 4 and (a + b)(1/a + 1/b) ≥ 4. Since all variables are
↪ positive, multiplying these two inequalities should preserve the inequality:

(x + y)(1/x + 1/y)(a + b)(1/a + 1/b) ≥ 4 * 4 = 16.

And that's exactly what I needed to prove! So, by applying the AM-GM inequality to each pair of
↪ terms separately and then multiplying the resulting inequalities, I've arrived at the desired
↪ conclusion.
```

```
theorem cauchy_p1 (x y a b : ℝ) (h₁ : a > 0 ∧ b > 0) (h₂ : x > 0 ∧ y > 0) : (
    x + y ) * ( 1 / x + 1 / y ) * ( a + b ) * ( 1 / a + 1 / b ) ≥ 16 := by
  have hx : x > 0 := h₂.1
  have hy : y > 0 := h₂.2
  have ha : a > 0 := h₁.1
  have hb : b > 0 := h₁.2
  have hxy : 0 < x * y := mul_pos hx hy
  have hab : 0 < a * b := mul_pos ha hb
  field_simp [hx.ne', hy.ne', ha.ne', hb.ne']
  rw [le_div_iff (by positivity), ← mul_assoc]
  nlinarith [sq_nonneg (x - y), sq_nonneg (a - b), sq_nonneg (x + y),
    sq_nonneg (a + b)]
```

### F.4   Example generations from Kimina-Distill-7B

Kimina-Distill-7B also first provide a thinking draft before generating the whole proof, and the formal
Lean 4 proof is aligned with the thinking draft. We omit the thinking draft here. Following is a
generation example from Kimina, which successfully decompose the problem into two sub-parts.

```
theorem cauchy_p16 (x y a b z w c d : ℝ) (hy : y ≠ 0) (hb : b ≠ 0) (hw : w ≠
    0) (hd : d ≠ 0) (hxy : x^2 + 1 / y^2 = 1) (hab : a^2 + 1 / b^2 = 4) (hzw :
    z^2 + 1 / w^2 = 1) (hcd : c^2 + 1 / d^2 = 4) : |(a / y + x / b) * (c / w +
    z / d)| ≤ 4 := by
  have h1 : (a / y + x / b) ^ 2 ≤ 4 := by
    have h1 : (a / y + x / b) ^ 2 ≤ (a ^ 2 + 1 / b ^ 2) * (x ^ 2 + 1 / y ^ 2)
    := by
      have h2 : (a ^ 2 + 1 / b ^ 2) * (x ^ 2 + 1 / y ^ 2) - (a / y + x / b) ^
    2 = (a * x - 1 / (b * y)) ^ 2 := by
        field_simp [hb, hy]
        ring
      have h3 : (a * x - 1 / (b * y)) ^ 2 ≥ 0 := by
        exact sq_nonneg (a * x - 1 / (b * y))
      linarith
    rw [hab, hxy] at h1
    nlinarith [sq_nonneg (a / y + x / b)]
  have h2 : (c / w + z / d) ^ 2 ≤ 4 := by
    have h1 : (c / w + z / d) ^ 2 ≤ (c ^ 2 + 1 / d ^ 2) * (z ^ 2 + 1 / w ^ 2)
    := by
      have h2 : (c ^ 2 + 1 / d ^ 2) * (z ^ 2 + 1 / w ^ 2) - (c / w + z / d) ^
    2 = (c * z - 1 / (d * w)) ^ 2 := by
        field_simp [hd, hw]
        ring
      have h3 : (c * z - 1 / (d * w)) ^ 2 ≥ 0 := by
        exact sq_nonneg (c * z - 1 / (d * w))
      linarith
    rw [hcd, hzw] at h1
```

```
      nlinarith [sq_nonneg (c / w + z / d)]
  have h3 : |(a / y + x / b)| ≤ 2 := by
    have h3 : -2 ≤ (a / y + x / b) := by
      nlinarith [sq_nonneg (a / y + x / b - 2), h1]
    have h4 : (a / y + x / b) ≤ 2 := by
      nlinarith [sq_nonneg (a / y + x / b - 2), h1]
    apply abs_le.mpr
    constructor <;> linarith
  have h4 : |(c / w + z / d)| ≤ 2 := by
    have h3 : -2 ≤ (c / w + z / d) := by
      nlinarith [sq_nonneg (c / w + z / d - 2), h2]
    have h4 : (c / w + z / d) ≤ 2 := by
      nlinarith [sq_nonneg (c / w + z / d - 2), h2]
    apply abs_le.mpr
    constructor <;> linarith
  have h5 : |(a / y + x / b) * (c / w + z / d)| ≤ 4 := by
    have h1 : -4 ≤ (a / y + x / b) * (c / w + z / d) := by
      nlinarith [sq_nonneg ((a / y + x / b) * (c / w + z / d) + 4), h3, h4]
    have h2 : (a / y + x / b) * (c / w + z / d) ≤ 4 := by
      nlinarith [sq_nonneg ((a / y + x / b) * (c / w + z / d) - 4), h3, h4]
    apply abs_le.mpr
    constructor <;> linarith
  exact h5
```

Following is another example where Kimina breaks the problem into two sub-parts.

```
theorem cauchy_p26 (x y a b : ℝ) (h₁ : a > 0 ∧ b > 0) (h₂ : x > 0 ∧ y > 0) : (
    x + y ) * ( 1 / x + 1 / y ) * ( a + b ) * ( 1 / a + 1 / b ) ≥ 16 := by
  have h1 : (x + y) * (1 / x + 1 / y) ≥ 4 := by
    have hxy1 : x > 0 := h₂.left
    have hy1 : y > 0 := h₂.right
    have h1 : (x + y) * (1 / x + 1 / y) - 4 = (x - y) ^ 2 / (x * y) := by
      field_simp
      ring
    have h2 : (x - y) ^ 2 ≥ 0 := by
      exact sq_nonneg (x - y)
    have h3 : x * y > 0 := mul_pos hxy1 hy1
    have h4 : (x - y) ^ 2 / (x * y) ≥ 0 := by
      apply div_nonneg
      · exact h2
      · linarith [h3]
    linarith [h1, h4]
  have h2 : (a + b) * (1 / a + 1 / b) ≥ 4 := by
    have ha1 : a > 0 := h₁.left
    have hb1 : b > 0 := h₁.right
    have h1 : (a + b) * (1 / a + 1 / b) - 4 = (a - b) ^ 2 / (a * b) := by
      field_simp
      ring
    have h2 : (a - b) ^ 2 ≥ 0 := by
      exact sq_nonneg (a - b)
    have h3 : a * b > 0 := mul_pos ha1 hb1
    have h4 : (a - b) ^ 2 / (a * b) ≥ 0 := by
      apply div_nonneg
      · exact h2
      · linarith [h3]
    linarith [h1, h4]
  nlinarith [h1, h2, mul_nonneg (le_of_lt h₁.left) (le_of_lt h₁.right),
    mul_nonneg (le_of_lt h₂.left) (le_of_lt h₂.right)]
```

Kimina-Distill-7B still sometimes treats the problem as a whole, even if it is not so clear how to use sum-of-square to prove the statement.

```
theorem amgm_p49 (a b c x y z: ℝ) (ap : a > 0) (bp : b> 0) (cp : c> 0) (xp : x
    > 0) (yp : y> 0) (zp : z> 0) : (a^3 + b^3 + c^3) * (x^3 + y^3 + z^3) ≥
    (a^2 * b + b^2 * c + c^2 * a) * (x^2 * y + y^2 * z + z^2 * x) := by
  nlinarith [sq_nonneg (a * x - b * y), sq_nonneg (a * y - b * z), sq_nonneg
    (a * z - b * x),
    sq_nonneg (b * x - c * y), sq_nonneg (b * y - c * z), sq_nonneg (b * z - c *
     x),
    sq_nonneg (c * x - a * y), sq_nonneg (c * y - a * z), sq_nonneg (c * z - a *
     x),
    sq_nonneg (a - b), sq_nonneg (b - c), sq_nonneg (c - a),
    sq_nonneg (x - y), sq_nonneg (y - z), sq_nonneg (z - x),
    mul_pos ap xp, mul_pos ap yp, mul_pos ap zp, mul_pos bp xp, mul_pos bp yp,
    mul_pos bp zp, mul_pos cp xp, mul_pos cp yp, mul_pos cp zp]
```

In the following example, the model explicitly uses the weighted AM-GM inequality with three variables from Mathlib (Real.geom_mean_le_arith_mean3_weighted). Note that Kimina model is the only model we tested that can correctly apply AM-GM inequality in Mathlib.

When successfully applying AM-GM, Kimina-Distill-7B can prove the statement easily.

```
theorem amgm_p6 (x y z: ℝ) (hx : x > 0) (hy : y > 0) (hz : z > 0) : (2:ℝ)/5 *
    x + (2:ℝ)/5 * y + (1:ℝ)/5 * z ≥ x ^ ((2:ℝ)/5) * y ^ ((2:ℝ)/5) * z ^
    ((1:ℝ)/5) := by
  have hx' : (0:ℝ) ≤ x := by linarith
  have hy' : (0:ℝ) ≤ y := by linarith
  have hz' : (0:ℝ) ≤ z := by linarith
  have hw₁ : (0:ℝ) ≤ (2:ℝ)/5 := by norm_num
  have hw₂ : (0:ℝ) ≤ (2:ℝ)/5 := by norm_num
  have hw₃ : (0:ℝ) ≤ (1:ℝ)/5 := by norm_num
  have hw : (2:ℝ)/5 + (2:ℝ)/5 + (1:ℝ)/5 = (1:ℝ) := by norm_num
  have h3 : (2:ℝ)/5 * x + (2:ℝ)/5 * y + (1:ℝ)/5 * z ≥ x ^ ((2:ℝ)/5) * y ^
    ((2:ℝ)/5) * z ^ ((1:ℝ)/5) := by
    apply Real.geom_mean_le_arith_mean3_weighted
    all_goals linarith
  exact h3
```

### F.5   Example generations from DeepSeek-Prover-V2-7B

DeepSeek-Prover-V2-7B generates code with very similar style to that of DeepSeek-R1-Distill-Qwen-32B, but has much stronger ability. Following is an example from Ineq-Mix that DeepSeek-Prover-V2-7B successfully solves. The problem is constructed by first composing two problems (Table 4) and then applying an algebraic transformation on both the LHS and RHS of the inequality. We can see that DeepSeek-Prover-V2-7B generates very structured code, and decomposes the problem into sub-parts.

```
theorem alge_whole_p70 (a b c : ℝ) ( ha : a > 0 ) ( hb : b > 0 ) ( hc : c > 0
    ) ( h0 : a + 2 * b = 3 ): ((3:ℝ) * (1) + (2:ℝ) * (( a + b + c ) * ( 1 / a +
    1 / b + 1 / c )))^3 ≥ ((3:ℝ) * (a * b^2) + (2:ℝ) * (9))^3 := by
  have h_main_inequality : (3:ℝ) + 2 * (a + b + c) * (1 / a + 1 / b + 1 / c) ≥
    21 := by
    have h₁ : 0 < a * b := by positivity
    have h₂ : 0 < a * c := by positivity
    have h₃ : 0 < b * c := by positivity
    have h₄ : 0 < a * b * c := by positivity
    have h₅ : 0 < a * b * c * a := by positivity
    have h₆ : 0 < a * b * c * b := by positivity
    have h₇ : 0 < a * b * c * c := by positivity
    field_simp
    rw [le_div_iff (by positivity)]
    nlinarith [sq_nonneg (a - b), sq_nonneg (a - c), sq_nonneg (b - c),
      sq_nonneg (a - 1), sq_nonneg (b - 1), sq_nonneg (c - 1)]
```

```
have h_rhs_bound : ((3:ℝ) * (a * b^2) + (2:ℝ) * (9)) ≤ 21 := by
  have h₁ : a + 2 * b = 3 := h0
  have h₂ : 0 < a := ha
  have h₃ : 0 < b := hb
  have h₄ : 0 < a * b^2 := by positivity
  have h₅ : 0 < a * b := by positivity
  -- Use non-linear arithmetic to prove the inequality
  nlinarith [sq_nonneg (b - 1), sq_nonneg (a - 1), mul_nonneg h₂.le
  (sq_nonneg (b - 1)),
    mul_nonneg h₃.le (sq_nonneg (a - 1)), mul_nonneg h₂.le (sq_nonneg (a -
  1)),
    mul_nonneg h₃.le (sq_nonneg (b - 1)), sq_nonneg (a - 3 + 2 * b),
    sq_nonneg (b - 3 / 2)]

have h_final : ((3:ℝ) * (1) + (2:ℝ) * (( a + b + c ) * ( 1 / a + 1 / b + 1 /
  c )))^3 ≥ ((3:ℝ) * (a * b^2) + (2:ℝ) * (9))^3 := by
  have h₁ : (3:ℝ) + 2 * (a + b + c) * (1 / a + 1 / b + 1 / c) ≥ 21 :=
  h_main_inequality
  have h₂ : ((3:ℝ) * (1) + (2:ℝ) * (( a + b + c ) * ( 1 / a + 1 / b + 1 / c
  ))) ≥ 21 := by
    linarith
  have h₃ : ((3:ℝ) * (a * b^2) + (2:ℝ) * (9)) ≤ 21 := h_rhs_bound
  have h₄ : ((3:ℝ) * (1) + (2:ℝ) * (( a + b + c ) * ( 1 / a + 1 / b + 1 / c
  ))) ≥ ((3:ℝ) * (a * b^2) + (2:ℝ) * (9)) := by
    nlinarith
  have h₅ : ((3:ℝ) * (1) + (2:ℝ) * (( a + b + c ) * ( 1 / a + 1 / b + 1 / c
  )))^3 ≥ ((3:ℝ) * (a * b^2) + (2:ℝ) * (9))^3 := by
    exact pow_le_pow_of_le_left (by
      -- Prove that the LHS is non-negative
      have h₆ : (3:ℝ) * (1) + (2:ℝ) * (( a + b + c ) * ( 1 / a + 1 / b + 1 /
  c )) ≥ 0 := by
        have h₇ : (a + b + c) * (1 / a + 1 / b + 1 / c) ≥ 0 := by positivity
        nlinarith
      nlinarith) h₄ 3
  exact h₅

exact h_final
```