# OpenReview forum: "Ineq-Comp: Benchmarking Human-Intuitive Compositional Reasoning in Automated Theorem Proving of Inequalities"
_NeurIPS.cc/2025/Datasets_and_Benchmarks_Track — NeurIPS 2025 Datasets and Benchmarks Track poster_

### Official Review · Reviewer_txmS · 2025-06-29

**Rating:** 5
**Confidence:** 5

**Summary:**

This paper evaluates the compositional reasoning abilities of LLM-based formal proof assistants using the Ineq-Comp benchmark, which tests provers on algebraic inequalities through human-intuitive transformations like variable duplication and algebraic rewrites. The results show that most models, including Goedel, STP, and Kimina-7B, struggle significantly with these simple transformations, revealing a gap in their compositional reasoning capabilities. Even when provided with formal proofs of the basic problems, these models fail to generalize effectively to transformed problems.

**Dataset Code Accessibility:**

Yes

**Dataset Code Comments:**

They have done an excellent job addressing the challenges related to code/data access, reproducibility, and completeness. Their approach ensures seamless access to well-documented code and metadata, which enhances transparency and facilitates easy replication of experiments. The inclusion of clear versioning and standardized documentation has significantly reduced discrepancies and barriers in reproducing results. Furthermore, they have made substantial efforts in curating and preprocessing data, ensuring that it is both complete and aligned with the models. This attention to detail not only promotes reproducibility but also sets a high standard for collaborative research and the sharing of resources.

**Ethical Considerations:**

No, there are no or only very minor ethics concerns

**Final Justification:**

Thank you to the author for answering my question. I will keep my score.

**Limitations Weaknesses:**

- Why doesn't the paper provide methods to improve compositional generalization?
- The results regarding Gemini and the GPT family are not provided.

**Strengths Contributions:**

- The paper is very easy to understand.
- This paper explores the issue of compositional generalization in theorem proving, which is very interesting.
- The paper provides detailed ablation experiments on the combination of different subgoals in theorem proving, enhancing the understanding of the model's compositional reasoning ability.

---

> ### Author Rebuttal · Authors · 2025-07-30
>
> Thank you very much for your time and effort to review our paper and the code/dataset. Below we answer your questions.
>
> > The results regarding Gemini and the GPT family are not provided.
>
> Thanks for raising this question. We indeed forgot to explicitly mention why we do not include these proprietary models directly on our main benchmark (AMGM/Cauchy/MISC-Seed/TypeI/TypeII). We actually have lightweight tests on GPT-4o, Gemini-2.5-flash (w/ thinking), Claude-3.7-Sonnet under pass@8, but the performances are all close to 0. We state the reasons below, and we will add it to our revised version:
>
> + Unclear Lean version: There are nuances between different Lean versions that cannot be ignored, and it is unclear which Lean version these proprietary models are trained on. These models can generate very clear proof sketch (in natural language), but the Lean code generated are very likely to be wrong . For example, the GPT family would even generate proofs in a combination of Lean 3 and Lean 4 grammar.
> + Hallucination for theorems that do not exist: These proprietary models are not heavily trained on Lean 4, and thus, they're very likely to suffer from hallucinations. The proofs generated by these models often contain theorems/premises that do not exist. Although the "hallucinated" theorem name indicates that the models generally know how to solve the problem in natural language, the whole proofs cannot pass the compilation and thus are not considered solved in the theorem proving setting.
> + Very expensive inference: There is indeed probability that given the correct Lean version, the strongest models, like Gemini, we can generate correct proofs with a very high budget. However, the cost to run these experiments are extremely high.
>
> Due to these reasons, instead of directly doing large-scale testing on these proprietary models, we conduct experiments on these models under the ICL setting, where we directly provide the seed problems with the corresponding proof. We hope that with only minimal knowledge of Lean, these models generalize well to the Type I and II problems, and Table 2 summarizes our results on GPT and Claude. From the results, we can observe that these proprietary models also struggle with composition in formal theorem proving.
>
> Based on your suggestion, we also add Gemini-2.5-flash (under the ICL setting) with pass@8 and thinking turned on. Following is the result
>
> | AMGM-Type1 | AMGM-Type2 | Cauchy-Type1 | Cauchy-Type2 | MISC-Type1 | MISC-Type2 |
> | --- | --- | --- | --- | --- | --- |
> | 12\% | 32\% | 20\% | 12\% | 32\% | 56\% |
>
> The results for Gemini-2.5-flash (with thinking) align with the conclusion of our ICL experiment, which emphasize that simple compositions in theorem proving remain a challenge for proprietary models.
>
> Thank you again for this question. We will explicitly discuss the proprietary models (and why we do not include them in the main experiments) in our next version.
>
> > Why doesn't the paper provide methods to improve compositional generalization?
>
>
> Thank you for your question! Yes our original goal is to demonstrate that the lack of composition exists in formal theorem proving. We agree that ideally there will be methods to improve compositionality, however we gradually found that even these simple compositions are not easy to be learned in a lightweight manner, especially after conducting ablation study for the ICL and the fine-tuning experiments. We hypothesize that it needs much more efforts to gradually solve it (including data augmentation, data synthesis and other invented techniques such as the ones introduced in DeepSeek-Prover-V2) and it is out of the scope of this paper. Even DeepSeek-Prover-V2-671B model do not fully solve the simple composition, which has much better compositionality compared to prior works. Another evidence is that the recent open-source model, Goedel-Prover-V2, also has a clear compositional gap, even though the model's performance is better than DeepSeek-Prover-V2-671B in many standard benchmarks including PutnamBench. This reemphasizes that even the simple compositionality seems highly non-trivial to tackle, and probably need  gradual efforts. Please refer to the response to Reviewer 7t6s for the detailed experiment results for these large/new models on our benchmark.
> However, although no methods to solve compositionality is proposed in our paper, we believe that by providing a different perspective from existing popular theorem proving benchmarks, our work can advance the theorem proving community.
>
> Thank you again for your time and support of our work. Please let us know if you have any further questions.

---

> > ### Author Response · Authors · 2025-08-05
> > **Follow up on rebuttal**
> >
> > Dear reviewer txmS,
> >
> > Thank you very much for your time and effort in reviewing our paper and code. As the author discussion period is approaching, we are reaching out to know if our responses and newly added experiments address your concern. If you have further questions, we are also happy to answer.

---

### Official Review · Reviewer_VwmG · 2025-07-03

**Rating:** 4
**Confidence:** 4

**Summary:**

This paper introduces a new dataset Ineq-Comp that investigate LLM mathematical inequalities. The dataset is curated from 75 seed problems with AM-GM inequality, Cauchy-Schwarz, Jensen’s inequality, and others from math Olympiads, and each problem is paired with a human-written Lean 4 proof. The curation includes (1) simple transformations of seed into 2 variant types, and (2) more complex transformations by 3 rules. The dataset also includes 50 real-world inequalities. As a result, the dataset consists of (1) Ineq-Simp with 150 problems, (2) Ineq-Mix with a claimed unbounded set of problems yet trial with 100 problems, and (3) Ineq-Real with 50 problems.



Evaluation is conducted on general-purpose language models, whole-proof generation models, and tree-search methods. Results indicate that models lack compositional ability especially in formal language. Experiments with in-context learning show inferior improvements on the task. Fine-tuning shows effective in domain but fails to generalize to out-of-domain tasks.

**Dataset Code Accessibility:**

Yes

**Ethical Considerations:**

No, there are no or only very minor ethics concerns

**Final Justification:**

After considering the authors’ rebuttal, I maintain my rating of 4. The authors addressed my concern about Type I and Type II difficulties. Further explanations are needed in the manuscript for the insignificance of the curated data in in-context learning.

**Limitations Weaknesses:**

* The paper does not explicitly refer to an open-sourced link to this dataset and the codebase for the transformations.

* It says in the paper that Type II is supposed to be more challenging than Type I, could you explain why in the experimental results, models perform inferior in Type I than in Type II (e.g., in Table 1)?

* Why does the performance of DeepSeek-R1-Distill-Qwen-32B (w thinking) in Ineq-AMGM drop with in-context learning?

**Strengths Contributions:**

*  Present a new dataset of mathematical inequalities with compositionality.

* The dataset curation is simple yet able to create challenging problems.

* Extensive experiments are conducted. Findings are interesting to the community.

* The paper is well-organized and easy to follow.

* Comprehensive literature review.

---

> ### Author Rebuttal · Authors · 2025-07-30
>
> Thanks for your effort to review our paper and our code/dataset. Below, we will address your concerns.
>
> > The paper does not explicitly refer to an open-sourced link to this dataset and the codebase for the transformations.
>
> Thanks for your comment! We will add the link to our dataset (on HF) and code (on Github) in our next version.
>
> > It says in the paper that Type II is supposed to be more challenging than Type I, could you explain why in the experimental results, models perform inferior in Type I than in Type II (e.g., in Table 1)?
>
> Thanks for your question. First, we need to clarify that the division of Type I and Type II is merely based on the way we modify the seed problem, and we do not intend to imply that Type II is harder than Type I for theorem prover. They represent two types of composition that can not be directly compared, but we do believe that for human, Type I should be easier than Type II.
>
> As for the reason why models typically did better on Type II questions, we believe it might be related to the distribution of training data. For math problems, there are lots of problems that has some low-level trasformation. For example, proving that $a^2+b^2 >= 2ab$ and $a + b >= \sqrt{ab}$ (for positive $a,b$) are all very common. However for the current dataset, the problems that truly requires high-level planning, decomposition, and abstraction might be limited. Thus, Type II might be more "in-distribution" for the current theorem proving training data. Therefore, the models might behave worse compared with Type II since it might lack these high-level decomposition/abstraction capabilities, although for human it might be simpler (because human has these capabilities).
>
>
> > Why does the performance of DeepSeek-R1-Distill-Qwen-32B (w thinking) in Ineq-AMGM drop with in-context learning?
>
>
> Thanks for your question. First, the degradation is within the standard deviation reported in Table 1, and thus might just be noise. Second, we hypothesize that it is related to the forklore fact that reasoning models are not very good at instruction following. Thus, even though we provide the seed problem to the model in the prompt, it might actually harm the model's ability to get the correct solution. We find that DeepSeek-R1-Distill-Qwen-32B (w/ thinking) sometimes applies our given proof and sometimes default back to its original proof style, and sometimes it is not correctly copying everything we provided (and thus make some error).
>
> Thank you again for your time and effort. We hope our responses can answer your questions.

---

> > ### Author Response · Authors · 2025-08-05
> > **Follow up on rebuttal**
> >
> > Dear reviewer VwmG,
> >
> > Thank you again for your time and effort. As the author discussion period is approaching, we are reaching out to know if our responses answer your questions. If you have further questions, we are also happy to answer.

---

> > ### Comment · Reviewer_VwmG · 2025-08-08
> > **Response to authors**
> >
> > The author response answers most of my questions. I will keep my rating.

---

### Official Review · Reviewer_7t6s · 2025-07-05

**Rating:** 4
**Confidence:** 4

**Summary:**

This paper proposes Ineq-Comp, a new benchmark built from elementary inequalities with the primary goal of assessing LLMs' compositional reasoning capability. Ineq-Comp consists of 75 seed problems, 150 problems by performing two types of transformations over seed problems, and 50 real-world inequality problems. Extensive experimental evaluations and analyses show that nearly all theorem provers except for DeepSeek-Prover-V2-7B have dramatically performance drop on reasoning simple composition problems. Furthermore, fine-tuning does not help to generalize to problems with new seeds.  Another interesting finding is that, although theorem prover generates some compositional strategy in their comments, the generated formal proofs do not leverage such compositional strategy.

**Additional Feedback:**

Please refer to the comments on strengths and weaknesses.

**Dataset Code Accessibility:**

Yes

**Dataset Code Comments:**

Both the code and data are publicly available, and there are proper documentation to re-run evaluations of this work.

**Ethical Considerations:**

No, there are no or only very minor ethics concerns

**Final Justification:**

I generally remain positive after the authors response. The new results shared by the authors confirm my slight concern that the compositional gap will shrink once the model size increases, suggesting the observed phenomenon is serious for small models and becomes less of an issue for larger models. It is generally known that smaller models (e.g., 7B) are incapable of many things; in that sense, the findings in this work become less interesting although it is still worth being published, which is the reason why not giving a higher score.

**Limitations Weaknesses:**

- Compositional reasoning is universal capability beyond math problems; even exclusively for math, there are way more other kinds of problems other inequality. It is unclear whether the findings in this work are specific to inequality problems or actually generalize. It is likely there are few training data about inequality reasoning, and sum-of-squares related tactics are usually sufficient for existing in-equality training dataset.
- Only relatively small (i.e. within 32B) theorem prover models have been studied. Would larger models (say DeepSeek-Prover-V2-671B) behave differently? If so, the result shall be reported, since it suggests scaling help to resolve compositional reasoning, which is perhaps a more important conclusion.
- For any interesting theorem proving problems, domain-specific tactics are usually necessary. It is very likely that the theorem prover models do know compositional reasoning is needed (as hinted in their comments), however, there are no proper high-level tactics available.  Lean4 supports a flexible macro system and defining high-level tactics is a relatively easy task for domain experts. With access to proper "tools" or high-level tactics int this context, the model can easily tackle the observed issues. One evidence is that two recent works (already cited by the authors) successfully tackle IMO-level inequality challenges. Another "analogy" explaining the observed issue is that string manipulations are straightforward, but directly generating low-level assembly code to process strings can be very challenging.

**Strengths Contributions:**

- This paper presents an interesting math benchmark, Ineq-Comp, which is intuitive to humans but turns out to be challenging for theorem provers.
- The new benchmark generation is carefully controlled with systematic compositional transformations and representative set of seed problems. Particularly, the seed problems have a variety of distributions across AM-GM inequality, Cauchy-Schwarz inequality, Jensen's inquitiy, Schur's inequaity, and others. And compositional transformations are simple and straightforward.
- Extensive experimental evaluations are conducted -- three categories of models and methods are assessed; furthermore, to mitigate the compositional challenges, in-context learning and fine-tuning have been studied.
- Valuable insights are provided to diagnose root causes of observed failures -- existing theorem provers have overly reliance on low-level algebraic tactics, particularly sum-of-squares related.

---

> ### Author Rebuttal · Authors · 2025-07-30
>
> Thanks for your time and effort for reviewing our paper, dataset, and code. Below we present more experiments and answer your questions.
>
> > Only relatively small models are tested and should study more on the scaling behavior.
>
> Thanks for your insightful suggestions. Following your comment, we test DeepSeek-Prover-V2-671B pass@32 to compare with DeepSeek-Prover-V2-7B due to limited resource. We also test Goedel-Prover-V2-8B and Goedel-Prover-V2-32B, another two models from the same family with different sizes. The results are listed below.
>
>
> | Model | Pass | AMGM-Seed | AMGM-Type1 | AMGM-Type2 |
> | --- |--- |--- |--- |--- |
> | DS-Prover-V2-7B | 32 | 75.0\% | 58.6\% | 52.5\% |
> | DS-Prover-V2-671B | 32 | 92\% | 76\% | 84\% |
> | Goedel-Prover-V2-8B | 32 | 64\% | 52\% | 56\% |
> |  | 128 | 76\% | 60\% | 60\% |
> | Goedel-Prover-V2-32B | 32 | 100\% | 64\% | 88\% |
> |  | 128 | 100\% | 68\% | 92\% |
>
> | Model | Pass | Cauchy-Seed | Cauchy-Type1 | Cauchy-Type2 |
> | --- |--- |--- |--- |--- |
> | DS-Prover-V2-7B | 32 | 64.6\% | 33.0\% | 35.0\% |
> | DS-Prover-V2-671B | 32 | 88\% | 56\% | 68\% |
> | Goedel-Prover-V2-8B | 32 | 60\% | 28\% | 44\% |
> |  | 128 | 68\% | 40\% | 52\% |
> | Goedel-Prover-V2-32B | 32 | 92\% | 48\% | 72\% |
> |  | 128 | 92\% | 60\% | 72\% |
>
> | Model | Pass | MISC-Seed | MISC-Type1 | MISC-Type2 |
> | --- |--- |--- |--- |--- |
> | DS-Prover-V2-7B | 32 | 59.1\% | 49.3\% | 38.8\% |
> | DS-Prover-V2-671B | 32 | 84\% | 68\% | 60\% |
> | Goedel-Prover-V2-8B | 32 | 84\% | 24\% | 48\% |
> |  | 128 | 96\% | 52\% | 64\% |
> | Goedel-Prover-V2-32B | 32 | 92\% | 72\% | 76\% |
> |  | 128 | 92\% | 76\% | 84\% |
>
>
>
> From the experiment results on larger and latest models, we can observe that
> + For larger models in the same series, performance on all benchmarks (AMGM/Cauchy/MISC-Seed/Type I/Type II) increases, which aligns with the intuition that larger model is indeed better.
> + On the contrary, the compositional gap of larger models, i.e., the performance gap between Seed and Type I or Type II, might not improve. For example, even if there are cases where the gap greatly decreses, on AMGM, Goedel-Prover-V2-32B reaches 100\% on seed problems, increasing \~30\% for both pass@32 and pass@128, while its Type I performance only slightly increases. Although it is arguable whether the gap is the exact measure for the true compositional gap, the expriements show that compositionality might not be easily solvable by merely scaling up.
>
> + Also, the results reinforce that our benchmark offers a different perspective compared with current benchmarks. For example, while Goedel-Prover-V2-8B outperforms DeepSeek-Prover-V2-671B on the most popular MiniF2F test set, its compositional gap is similar to DeepSeek-Prover-V2-7B, and surprisingly larger than 671B.
>
> In short, even with scaling up, the compositional gap still remains. We believe that the new experiments will enhance the clarity and impact of our paper.
>
> > Compositional reasoning is universal capability beyond math problems; even exclusively for math, there are way more other kinds of problems other inequality. It is unclear whether the findings in this work are specific to inequality problems or actually generalize. It is likely there are few training data about inequality reasoning, and sum-of-squares related tactics are usually sufficient for existing inequality training dataset.
>
>
> We agree compositional reasoning is a universal capability. However, we want to emphasize that our paper aims at theorem proving in formal language, instead of general math problems. We chose inequalities as a clean, controllable testbed because they allow us to start from seed proofs and apply minimal, human-intuitive transformations (e.g., duplicating variables, simple rewrites) that expose compositional failures, while compositional problems in other fields are generally difficult to systematically construct and test in a controllable manner. Although we do not test other domains, compositional reasoning is core to multi-step problem solving everywhere, and we believe the deficiency highlighted here is likely shared across domains.
>
> We acknowledge that data scarcity might be a factor contributing to the poor performance of the models. In fact, this is exactly what we tried to point out in line 85-89. However, we must emphasize that data scarcity of inequality problems alone cannot fully explain the observed limitations. If theorem provers genuinely have the ability to abstract, decompose, and reuse results in a human-like manner, they shouldn't exhibit the significant performance gap between seed problems and Type I problems. That is to say, with compositional abilities, even if the models only know how to solve inequalities using sum-of-squares, ideally they should be capable of solving Type I problems (or even Type II problems) by using proper compositions on sum-of-squares. Therefore, we claim that our work effectively captures the compositional reasoning deficiency of current models.
>
>
> > For any interesting theorem proving problems, domain-specific tactics are usually necessary. It is very likely that the theorem prover models do know compositional reasoning is needed (as hinted in their comments), however, there are no proper high-level tactics available. Lean4 supports a flexible macro system and defining high-level tactics is a relatively easy task for domain experts. With access to proper "tools" or high-level tactics int this context, the model can easily tackle the observed issues. One evidence is that two recent works (already cited by the authors) successfully tackle IMO-level inequality challenges. Another "analogy" explaining the observed issue is that string manipulations are straightforward, but directly generating low-level assembly code to process strings can be very challenging.
>
>
> Thanks for your comment. We agree that domain-specific tactics can help, but our benchmark is designed so that at least Type I problems can be solved without any new tactics by human: abstracting the seed lemma, simply calling the seed lemma twice, and applying `nlinarith` (with some very straightforward proof for positivity) suffices. In other words, the difficulty is not from missing tactics but from the model’s inability to decompose, abstract, and reuse known results.
>
> As we noted, models sometimes indicate in comments that composition is needed, but this informal recognition does not translate into formal proof construction. We view this gap as precisely a lack of compositional ability since we focus on formal theorem proving. In the “string analogy,” generating low-level assembly to implement higher-level operations is itself a form of compositional reasoning required in formal theorem proving. Even with flexible macro systems, provers must still learn to use existing tools correctly, which remains a key bottleneck as is shown by our results.
>
> While domain-specific systems like AIPS and Li et al. 2025 succeed on inequalities by hand-crafting tactics (note that they also have limitations, e.g., AIPS can only solve 3 or 4 variables cyclic inequalities that is very in-distribution for math contest), our benchmark targets general-purpose provers, which need to tackle out-of-distribution problems across domains, similar to how LLMs aim for broad generalization. It is impractical to build custom tactics for every mathematical subfield, and ultimately, solving truly hard and novel problems requires models to flexibly combine knowledge across domains. This is why compositional ability in formal reasoning remains essential, even when high-level tactics are available.
>
> Thank you again for your time and effort to review our paper and give constructive suggestions, and we hope that our new experiments and our responses can address your concern.

---

> > ### Author Response · Authors · 2025-08-05
> > **Follow up on rebuttal**
> >
> > Dear reviewer 7t6s,
> >
> > Thank you again for your time and effort in reviewing our paper and code. As the author discussion period is approaching, we are reaching out to know if our responses and newly added experiments address your concern. If you have further questions, we are also happy to answer.

---

### Decision · Program_Chairs · 2025-09-18

**Decision:**

Accept (poster)

**Comment:**

The paper creates a new benchmark for proving math inequalities and studies different families of models on those benchmarks. The paper is well-written, and clearly highlights the main challenges in this fields for current LLM. Even for simple problems, the models often struggle to provide a clean Lean 4 code for the proof.  I think it is a useful contribution.